# On-Policy Policy Gradient Reinforcement Learning Without On-Policy Sampling

## Abstract

On-policy reinforcement learning (RL) algorithms perform policy updates using i.i.d. trajectories collected by the current policy. However, after observing only a finite number of trajectories, on-policy sampling may produce data that fails to match the expected on-policy data distribution. This *sampling error* leads to noisy updates and data inefficient on-policy learning. Recent work in the policy evaluation setting has shown that non-i.i.d., off-policy sampling can produce data with lower sampling error than on-policy sampling can produce (Zhong et al., 2022). Motivated by this observation, we introduce an adaptive, off-policy sampling method to improve the data efficiency of on-policy policy gradient algorithms. Our method, **P**roximal **R**obust **O**n-**P**olicy **S**ampling (PROPS), reduces sampling error by collecting data with a *behavior policy* that increases the probability of sampling actions that are under-sampled with respect to the current policy. We empirically evaluate PROPS on both continuous-action MuJoCo benchmark tasks as well discrete-action tasks and demonstrate that (1) PROPS decreases sampling error throughout training and (2) improves the data efficiency of on-policy policy gradient algorithms.

## 1 Introduction

One of the most widely used classes of reinforcement learning (RL) algorithms is the class of on-policy policy gradient algorithms. These algorithms use gradient ascent on the parameters of a parameterized policy so as to increase the probability of observed actions with high expected returns under the current policy. The gradient is commonly estimated using the Monte Carlo estimator, an average computed over i.i.d. samples of trajectories from the current policy. The Monte Carlo estimator is consistent and unbiased; as the number of sampled trajectories increases, the empirical distribution of trajectories converges to the true distribution under the current policy, and thus the empirical gradient converges to the true gradient. However, the expense of environment interactions forces us to work with finite samples. Thus, the empirical distribution of the trajectories often differs from the desired on-policy data distribution. We refer to the mismatch between the empirical distribution of trajectories and the desired on-policy trajectory distribution as *sampling error*. This sampling error produces inaccurate gradient estimates, resulting in noisy policy updates, slower learning, and potentially convergence to suboptimal policies. With i.i.d. on-policy sampling, the only way to reduce sampling error is to collect more data.

Since on-policy sampling is so widely used to produce on-policy data, on-policy sampling is often taken to be an essential feature of data collection for on-policy learning (Silver, 2015; Achiam, 2018; Sutton and Barto, 2018). However, on-policy sampling is not explicitly required for on-policy learning; on-policy learning requires on-policy *data* – data whose state-conditioned empirical distribution of actions matches that of the current policy. On-policy sampling is a straightforward way to acquire on-policy data, though we can obtain such data more efficiently *without* on-policy sampling. To better illustrate this concept, consider an MDP with two discrete actions A and B, and suppose the current policy $\pi$ places equal probability on both actions in some state $s$. When following $\pi$, after 10 visits to $s$, we may observe A 2 times and B 8 times rather than the expected 5 times. Alternatively, if we adaptively select the most under-sampled action upon every visit to $s$, we will observe each action an equal number of times. The first scenario illustrates on-policy sampling but not on-policy data; the second scenario uses *off-policy* sampling yet produces on-policy data.

Figure 1: An overview of PROPS for on-policy policy gradient learning. Rather than collecting data $\mathcal{D}$ via on-policy sampling from the agent's current policy $\pi_{\boldsymbol{\theta}}$, we collect data with a separate data collection policy $\pi_{\phi}$ that we continually adapt to reduce sampling error in $\mathcal{D}$ with respect to the agent's current policy.

These observations raise the following question: can on-policy policy gradient algorithms learn more efficiently using on-policy data acquired *without* on-policy sampling? Recently, Zhong et al. (2022) showed that adaptive, off-policy sampling can yield data that more closely matches the on-policy distribution than data produced by i.i.d. on-policy sampling. However, this work was limited to the policy evaluation setting in which the on-policy distribution remains fixed. Turning from evaluation to control poses the challenge of a continually changing current policy.

In this work, we address this challenge and show for the first time that on-policy policy gradient algorithms are more data-efficient learners when they use on-policy data acquired with adaptive, off-policy sampling. Our method, **P**roximal **R**obust **O**n-**P**olicy **S**ampling (PROPS)[1], adaptively corrects sampling error in previously collected data by increasing the probability of sampling actions that are under-sampled with respect to the current policy. Fig. 1 provides an overview of PROPS. We empirically evaluate PROPS on continuous-action MuJoCo benchmark tasks as well as discrete action tasks and show that (1) PROPS reduces sampling error throughout training and (2) improves the data efficiency of on-policy policy gradient algorithms. In summary, our contributions are

1. We introduce an adaptive sampling algorithm that reduces sampling error in on-policy data collection.

2. We demonstrate empirically that our method improves the data efficiency of on-policy policy gradient algorithms and increases the fraction of training runs that converge to high-return polices.

3. Building off of the theoretical foundation laid by Zhong et al. (2022), this work improves the RL community's understanding of a nuance in the on-policy vs off-policy dichotomy: on-policy learning requires on-policy data, not on-policy sampling.

## 2 RELATED WORK

Our work focuses on data collection in RL. In RL, data collection is often framed as an exploration problem, focusing on how an agent should explore its environment to efficiently learn an optimal policy. Prior RL works have proposed several exploration-promoting methods such as intrinsic motivation (Pathak et al., 2017; Sukhbaatar et al., 2018), count-based exploration (Tang et al., 2017; Ostrovski et al., 2017), and Thompson sampling (Osband et al., 2013; Sutton and Barto, 2018). In contrast, our objective is to learn from the on-policy data distribution; we use adaptive data collection to more efficiently obtain this data distribution.

Prior works have used adaptive off-policy sampling to reduce sampling error in the policy evaluation subfield of RL. Most closely related is the work of Zhong et al. (2022) who first proposed that adaptive off-policy sampling could produce data that more closely matches the on-policy distribution than on-policy sampling could produce. Mukherjee et al. (2022) use a deterministic sampling rule to take actions in a particular proportion. Other bandit works use a non-adapative exploration policy to collect additional data conditioned on previously collected data (Tucker and Joachims, 2022; Wan

---

[1]We include our codebase in the supplemental material.

et al., 2022; Konyushova et al., 2021). Since these works only focus on policy evaluation, they do not have to contend with a changing on-policy distribution as our work does for the control setting.

Several prior works propose importance sampling methods (Precup, 2000) to reduce sampling error without further data collection. In the RL setting, Hanna et al. (2021) showed that reweighting off-policy data according to an estimated behavior policy can correct sampling error and improve policy evaluation. Similar methods have been studied for temporal difference learning (Pavse et al., 2020) and policy evaluation in the bandit setting (Li et al., 2015; Narita et al., 2019). Conservative Data Sharing (Yu et al., 2021) reduces sampling error by selectively integrating offline data from multiple tasks. Our work instead focuses on using additional data collection to reduce sampling error.

As we will discuss in Section 5, the method we introduce permits data collected in one iteration of policy optimization to be re-used in future iterations rather than discarded as typically done by on-policy algorithms. Prior work has attempted to avoid discarding data by combining off-policy and on-policy updates with separate loss functions or by using alternative gradient estimates (Wang et al., 2016; Gu et al., 2016; 2017; Fakoor et al., 2020; O'Donoghue et al., 2016; Queeney et al., 2021). In contrast, our method modifies the sampling distribution at each iteration so that the entire data set of past and newly collected data matches the expected distribution under the current policy.

## 3 PRELIMINARIES

### 3.1 REINFORCEMENT LEARNING

We formalize the RL environment as a finite horizon Markov decision process (MDP) (Puterman, 2014) $(\mathcal{S}, \mathcal{A}, p, r, d_0, \gamma)$ with state space $\mathcal{S}$, action space $\mathcal{A}$, transition dynamics $p : \mathcal{S} \times \mathcal{A} \times \mathcal{S} \to [0, 1]$, reward function $r : \mathcal{S} \times \mathcal{A} \to \mathbb{R}$, initial state distribution $d_0$, and reward discount factor $\gamma \in [0, 1)$. The state and action spaces may be discrete or continuous. We write $p(\cdot \mid \boldsymbol{s}, \boldsymbol{a})$ to denote the distribution of next states after taking action $\boldsymbol{a}$ in state $\boldsymbol{s}$. We consider stochastic policies $\pi_{\boldsymbol{\theta}} : \mathcal{S} \times \mathcal{A} \to [0, 1]$ parameterized by $\boldsymbol{\theta}$, and we write $\pi_{\boldsymbol{\theta}}(\boldsymbol{a}|\boldsymbol{s})$ to denote the probability of sampling action $\boldsymbol{a}$ in state $\boldsymbol{s}$ and $\pi_{\boldsymbol{\theta}}(\cdot|\boldsymbol{s})$ to denote the probability distribution over actions in state $\boldsymbol{s}$. We additionally let $d_{\pi_{\boldsymbol{\theta}}} : \mathcal{S} \times \mathcal{A} \to [0, 1]$ denote the state-action visitation distribution, the distribution over state-action pairs induced by following $\pi_{\boldsymbol{\theta}}$. The RL objective is to find a policy that maximizes the expected sum of discounted rewards, defined as:

$$J(\boldsymbol{\theta}) = \mathbb{E}_{\tau \sim \pi_{\boldsymbol{\theta}}} \left[ \sum_{t=0}^{H} \gamma^t r(\boldsymbol{s}_t, \boldsymbol{a}_t) \right], \tag{1}$$

where the horizon $H$ is the random variable representing the time-step when an episode ends. Throughout this paper, we refer to the policy used for data collection as the *behavior policy* and the policy trained to maximize its expected return as the *target policy*.

### 3.2 ON-POLICY POLICY GRADIENT ALGORITHMS

Policy gradient algorithms are one of the most widely used methods in RL. These methods perform gradient ascent over policy parameters to maximize an agent's expected return $J(\boldsymbol{\theta})$ (Eq. 1). The gradient of the $J(\boldsymbol{\theta})$ with respect to $\boldsymbol{\theta}$, or *policy gradient*, is often given as:

$$\nabla_{\boldsymbol{\theta}} J(\boldsymbol{\theta}) = \mathbb{E}_{\boldsymbol{s} \sim d_{\pi_{\boldsymbol{\theta}}}^{\gamma}, \boldsymbol{a} \sim \pi_{\boldsymbol{\theta}}} \left[ A^{\pi_{\boldsymbol{\theta}}}(\boldsymbol{s}, \boldsymbol{a}) \nabla_{\boldsymbol{\theta}} \log \pi_{\boldsymbol{\theta}}(\boldsymbol{a}|\boldsymbol{s}) \right], \tag{2}$$

where $A^{\pi_{\boldsymbol{\theta}}}(\boldsymbol{s}, \boldsymbol{a})$ is the *advantage* of choosing action $\boldsymbol{a}$ in state $\boldsymbol{s}$ and following $\pi_{\boldsymbol{\theta}}$ thereafter. In practice, the expectation in Eq. 2 is approximated with Monte Carlo samples collected from $\pi_{\boldsymbol{\theta}}$ and an estimate of $A^{\pi_{\boldsymbol{\theta}}}$ used in place of the true advantages (Schulman et al., 2016). After updating the policy parameters with this estimated gradient, the previously collected trajectories $\mathcal{D}$ become off-policy with respect to the updated policy. To ensure gradient estimation remains unbiased, on-policy algorithms discard historic data after each update and collect new data with the updated policy.

This foundational idea of policy learning via stochastic gradient ascent was first proposed by Williams (Williams, 1992) under the name REINFORCE. Since then, a large body of research has focused on developing more scalable policy gradient methods (Kakade, 2001; Schulman et al., 2015; Mnih et al., 2016; Espeholt et al., 2018; Lillicrap et al., 2015; Haarnoja et al., 2018). Arguably, the

most successful variant of policy gradient learning is proximal policy optimization (PPO) (Schulman et al., 2017), the algorithm of choice in several high-profile success stories (Berner et al., 2019; Akkaya et al., 2019; Vinyals et al., 2019). Rather than maximizing the standard RL objective (Eq. 1), PPO maximizes a surrogate objective:

$$\mathcal{L}_{\text{PPO}}(\boldsymbol{s}, \boldsymbol{a}, \boldsymbol{\theta}, \boldsymbol{\theta}_{\text{old}}) = \min(g(\boldsymbol{s}, \boldsymbol{a}, \boldsymbol{\theta}, \boldsymbol{\theta}_{\text{old}}) A^{\pi_{\boldsymbol{\theta}_{\text{old}}}}(\boldsymbol{s}, \boldsymbol{a}),$$
$$\texttt{clip}(g(\boldsymbol{s}, \boldsymbol{a}, \boldsymbol{\theta}, \boldsymbol{\theta}_{\text{old}}), 1 - \epsilon, 1 + \epsilon) A^{\pi_{\boldsymbol{\theta}_{\text{old}}}}(\boldsymbol{s}, \boldsymbol{a})), \tag{3}$$

where $\boldsymbol{\theta}_{\text{old}}$ denotes the policy parameters prior to the update, $g(\boldsymbol{s}, \boldsymbol{a}, \boldsymbol{\theta}, \boldsymbol{\theta}_{\text{old}})$ is the policy ratio $g(\boldsymbol{s}, \boldsymbol{a}, \boldsymbol{\theta}, \boldsymbol{\theta}_{\text{old}}) = \frac{\pi_{\boldsymbol{\theta}}(\boldsymbol{a}|\boldsymbol{s})}{\pi_{\boldsymbol{\theta}_{\text{old}}}(\boldsymbol{a}|\boldsymbol{s})}$, and the clip function with hyperparameter $\epsilon$ clips $g(\boldsymbol{s}, \boldsymbol{a}, \boldsymbol{\theta}, \boldsymbol{\theta}_{\text{old}})$ to the interval $[1 - \epsilon, 1 + \epsilon]$. This objective disincentivizes large changes to $\pi_{\boldsymbol{\theta}}(\boldsymbol{a}|\boldsymbol{s})$. In contrast to other policy gradient algorithms which perform a single gradient update per data sample to avoid destructively large weight updates, PPO's clipping mechanism allows the agent to perform multiple epochs of minibatch policy updates.

## 4 CORRECTING SAMPLING ERROR IN REINFORCEMENT LEARNING

In this section, we illustrate how sampling error can produce inaccurate policy gradient estimates and then describe how adaptive, off-policy sampling can reduce sampling error. For exposition, we assume finite state and action spaces. The policy gradient can then be written as:

$$\nabla_{\boldsymbol{\theta}} J(\boldsymbol{\theta}) = \sum_{(\boldsymbol{s}, \boldsymbol{a}) \in \mathcal{S} \times \mathcal{A}} d_{\pi_{\boldsymbol{\theta}}}^{\gamma}(\boldsymbol{s}, \boldsymbol{a}) \left[ A^{\pi_{\boldsymbol{\theta}}}(\boldsymbol{s}, \boldsymbol{a}) \nabla_{\boldsymbol{\theta}} \log \pi_{\boldsymbol{\theta}}(\boldsymbol{a}|\boldsymbol{s}) \right]. \tag{4}$$

The policy gradient is thus a linear combination of the gradient for each $(\boldsymbol{s}, \boldsymbol{a})$ pair $\nabla_{\boldsymbol{\theta}} \log \pi_{\boldsymbol{\theta}}(\boldsymbol{a}|\boldsymbol{s})$ weighted by $d_{\pi_{\boldsymbol{\theta}}}^{\gamma}(\boldsymbol{s}, \boldsymbol{a}) A^{\pi_{\boldsymbol{\theta}}}(\boldsymbol{s}, \boldsymbol{a})$. Let $\mathcal{D}$ be a dataset of trajectories. It is straightforward to show that the Monte Carlo estimate of the policy gradient can be written in a similar form as Equation 4 except with the true state-action visitation distribution replaced with the empirical visitation distribution, $d_{\mathcal{D}}(\boldsymbol{s}, \boldsymbol{a})$ (Hanna et al., 2021). Consequently, when $(\boldsymbol{s}, \boldsymbol{a})$ is over-sampled (i.e., $d_{\mathcal{D}}(\boldsymbol{s}, \boldsymbol{a}) > d_{\pi_{\boldsymbol{\theta}}}^{\gamma}(\boldsymbol{s}, \boldsymbol{a})$), then $\nabla_{\boldsymbol{\theta}} \log \pi_{\boldsymbol{\theta}}(\boldsymbol{a}|\boldsymbol{s})$ contributes more to the overall gradient than it should. Similarly, when $(\boldsymbol{s}, \boldsymbol{a})$ is under-sampled, $\nabla_{\boldsymbol{\theta}} \log \pi_{\boldsymbol{\theta}}(\boldsymbol{a}|\boldsymbol{s})$ contributes less than it should.

We now provide a concrete example illustrating how small amounts of sampling error can cause the wrong actions to be reinforced, resulting in sub-optimal convergence. Suppose that in a particular state $\boldsymbol{s}_0$, an agent places equal probability on two actions $\boldsymbol{a}_0$ and $\boldsymbol{a}_1$ with advantages $A^{\pi_{\boldsymbol{\theta}}}(\boldsymbol{s}_0, \boldsymbol{a}_0) = 20$ and $A^{\pi_{\boldsymbol{\theta}}}(\boldsymbol{s}_0, \boldsymbol{a}_1) = 15$, respectively. Since $\nabla_{\boldsymbol{\theta}} \log \pi_{\boldsymbol{\theta}}(\boldsymbol{a}_0|\boldsymbol{s}_0) = \nabla_{\boldsymbol{\theta}} \log \pi_{\boldsymbol{\theta}}(\boldsymbol{a}_1|\boldsymbol{s}_0)$ and $d_{\pi_{\boldsymbol{\theta}}}^{\gamma}(\boldsymbol{s}_0, \boldsymbol{a}_0) = d_{\pi_{\boldsymbol{\theta}}}^{\gamma}(\boldsymbol{s}_0, \boldsymbol{a}_1)$, the expected gradient will increase the probability of sampling the action with the larger advantage ($\boldsymbol{a}_0$). With on-policy sampling, after 10 visits to $\boldsymbol{s}_0$, the agent will sample both actions 5 times in expectation. However, the agent may actually observe $\boldsymbol{a}_0$ 4 times and $\boldsymbol{a}_1$ 6 times. A Monte Carlo estimate of the policy gradient would then place $0.4 \cdot A^{\pi_{\boldsymbol{\theta}}}(\boldsymbol{s}_0, \boldsymbol{a}) = 8$ weight on the gradient of $\boldsymbol{a}_0$ and $0.6 \cdot A^{\pi_{\boldsymbol{\theta}}}(\boldsymbol{s}_0, \boldsymbol{a}_1) = 9$ weight on $\boldsymbol{a}_1$, thus *decreasing* the probability of sampling the optimal $\boldsymbol{a}_0$ action.

Sampling error in on-policy sampling vanishes as the size of the batch of data used to estimate the gradient tends toward infinity. However, the preceding example suggests a simple strategy that would eliminate sampling error with finite data: have the agent adapt its probability on the next action it takes based on what actions it has already sampled. Continuing with our example, suppose the agent has visited $\boldsymbol{s}_0$ 9 times and sampled $\boldsymbol{a}_0$ 4 times and $\boldsymbol{a}_1$ 5 times. With on-policy sampling, the agent may observe $\boldsymbol{a}_1$ again upon the next visit to $\boldsymbol{s}_0$. Alternatively, the agent could sample its next action from a distribution that puts probability 1 on $\boldsymbol{a}_0$ and consequently produce an aggregate batch of data that contains both actions in their expected frequency. While this adaptive method is an off-policy sampling method, it produces data that exactly matches the on-policy distribution and will thus produce a more accurate gradient.

This example suggests that we can heuristically reduce sampling error by taking the most under-sampled action at a given state. Under a strong assumption that the MDP had a DAG structure, Zhong et al. (2022) proved that this heuristic results in the empirical distribution of states and actions in a fixed-horizon MDP converging to $d_{\pi_{\boldsymbol{\theta}}}(\boldsymbol{s}, \boldsymbol{a})$ and moreover converging at a faster rate than on-policy sampling. We remove this limiting assumption with the following result:

**Proposition 1.** *Assume that data is collected with an adaptive behavior policy that always takes the most under-sampled action in each state, $s$, with respect to policy $\pi$, i.e, $a \leftarrow \arg\max_{a'}(\pi(a'|s) - \pi_{\mathcal{D}}(a'|s))$, where $\pi_{\mathcal{D}}$ is the empirical policy after $m$ state-action pairs have been collected. Assume that $\mathcal{S}$ and $\mathcal{A}$ are finite and that the Markov chain induced by $\pi$ is irreducible. Then we have that the empirical state visitation distribution, $d_m$, converges to the state distribution of $\pi$, $d_\pi$, with probability 1:*

$$\forall s, \lim_{m \to \infty} d_m(s) = d_\pi(s).$$

*Proof.* See Appendix A. □

While adaptively sampling the most under-sampled action can reduce sampling error, this heuristic is difficult to implement in practice. In tasks with continuous states and actions, the $\arg\max$ often has no closed-form solution, and the empirical policy can be expensive to compute at every timestep. Building upon the concepts discussed in this section, the following section presents a *scalable* adaptive sampling algorithm that reduces sampling error in on-policy policy gradient learning.

## 5 PROXIMAL ROBUST ON-POLICY SAMPLING FOR POLICY GRADIENT ALGORITHMS

Our goal is to develop an adaptive, off-policy sampling algorithm that reduces sampling error in on-policy data collection for on-policy policy gradient algorithms. We outline a general framework for on-policy learning with an adaptive behavior policy in Algorithm 1. In this framework, the behavior policy $\pi_\phi$ and target policy $\pi_\theta$ are initially the same. The behavior policy collects a batch of $m$ transitions, adds the batch to a data buffer $\mathcal{D}$, and then updates its weights such that the next batch it collects reduces sampling error in $\mathcal{D}$ with respect to the target policy $\pi_\theta$ (Lines 7-10). Every $n$ steps (with $n > m$), the agent updates its target policy with data from $\mathcal{D}$ (Line 11). We refer to $m$ and $n$ as the *behavior batch size* and the *target batch size*, respectively.

A subtle implication of adaptive sampling is that it can correct sampling error in *any* empirical data distribution – even one generated by a different policy. Rather than discarding off-policy data from old policies – as is commonly done in on-policy learning – we let the data buffer hold up to $b$ target batches ($bn$ transitions) and call $b$ the *buffer size*. If $b > 1$, then $\mathcal{D}$ will contain historic off-policy data used in previous target policy updates. Regardless of how $b$ is set, the role of the behavior policy is to continually adjust action probabilities for new samples so that the aggregate data distribution of $\mathcal{D}$ matches the expected on-policy distribution of the current target policy (Line 10). Implementing Line 10 is the core challenge we address in the remainder of this section.

---

**Algorithm 1** On-policy policy gradient algorithm with adaptive sampling

---

1: **Inputs**: Target batch size $n$, behavior batch size $m$, buffer size $b$.
2: **Output:** Target policy parameters $\boldsymbol{\theta}$.
3: Initialize target policy parameters $\boldsymbol{\theta}$.
4: Initialize behavior policy parameters $\boldsymbol{\phi} \leftarrow \boldsymbol{\theta}$.
5: Initialize empty buffer $\mathcal{D}$ with capacity $bn$.
6: **for** target update $i = 1, 2, \ldots$ **do**
7:     **for** behavior update $j = 1, \ldots, \lfloor n/m \rfloor$ **do**
8:         Collect batch of data $\mathcal{B}$ by running $\pi_\phi$.
9:         Append $\mathcal{B}$ to buffer $\mathcal{D}$.
10:         Update $\pi_\phi$ with $\mathcal{D}$ using Algorithm 2.
11:     Update $\pi_\theta$ with $\mathcal{D}$.
12: **return** $\boldsymbol{\theta}$

---

To ensure that the empirical distribution of $\mathcal{D}$ matches the expected on-policy distribution, updates to $\pi_\phi$ should attempt to increase the probability of actions which are currently under-sampled with respect to $\pi_\theta$. Zhong et al. (2022) recently developed a simple method called Robust On-policy Sampling (ROS) for making such updates. In particular, the gradient $\nabla_\phi \mathcal{L} := -\nabla_\phi \sum_{(\boldsymbol{s},\boldsymbol{a}) \in \mathcal{D}} \log \pi_\phi(\boldsymbol{a}|\boldsymbol{s})$ *when evaluated at $\phi = \theta$* provides a direction to change $\phi$ such that under-sampled actions have their probabilities increased. Thus a single step of gradient ascent will increase the probability of under-sampled actions.[2] In theory and in simple RL policy evaluation

---

[2]To add further intuition for this update, note that it is the opposite of a gradient ascent step on the log likelihood of $\mathcal{D}$. When starting at $\boldsymbol{\theta}$, gradient ascent on the data log likelihood will increase the probability of actions that are over-sampled relative to $\pi_\theta$. Hence, the ROS update changes $\phi$ in the opposite direction.

tasks, this update was shown to improve the rate at which the empirical data distribution converges to the on-policy distribution – even when the empirical data distribution contains off-policy data. Unfortunately, there are two main challenges that render ROS unsuitable for Line 10 in Algorithm 1.

**Challenge 1: Destructively large policy updates.** Since the buffer $\mathcal{D}$ may contain data collected from older target policies, some samples in $\mathcal{D}$ may be very off-policy with respect to the current target policy such that $\log \pi_\phi(\boldsymbol{a}|\boldsymbol{s})$ is large and negative. Since $\nabla_\phi \log \pi_\phi(\boldsymbol{a}|\boldsymbol{s})$ increases in magnitude as $\pi_\phi(\boldsymbol{a}|\boldsymbol{s})$ tends towards zero, ROS incentivizes the agent to continually decrease the probability of these actions despite being extremely unlikely under the current target policy. Thus, off-policy samples can produce destructively large policy updates.

**Challenge 2: Improper handling of continuous actions.** In a continuous-action task, ROS may produce behavior policies that *increase* sampling error. A continuous-action task policy $\pi_\theta(\boldsymbol{a}|\boldsymbol{s})$ is

---

**Algorithm 2** PROPS Update

1: **Inputs:** Target policy parameters $\boldsymbol{\theta}$, buffer $\mathcal{D}$, target KL $\delta$, clipping coefficient $\epsilon_{\text{PROPS}}$, regularizer coefficient $\lambda$, n_epoch, n_minibatch.
2: **Output:** Behavior policy parameters $\boldsymbol{\phi}$.
3: $\boldsymbol{\phi} \leftarrow \boldsymbol{\theta}$
4: **for** epoch $i = 1, 2, \ldots,$ n_epoch **do**
5:     **for** minibatch $j = 1, 2, \ldots,$ n_minibatch **do**
6:         Sample minibatch $\mathcal{D}_j \sim \mathcal{D}$
7:         Compute the loss (Eq. 6)

$$\mathcal{L} \leftarrow \frac{1}{|\mathcal{D}_j|} \sum_{(\boldsymbol{s},\boldsymbol{a}) \in \mathcal{D}_j} \mathcal{L}_{\text{PROPS}}(\boldsymbol{s}, \boldsymbol{a}, \boldsymbol{\phi}, \boldsymbol{\theta}, \epsilon_{\text{PROPS}}, \lambda)$$

8:         Update $\boldsymbol{\phi}$ with a step of gradient ascent on $\mathcal{L}$
9:         **if** $D_{\text{KL}}(\pi_\theta || \pi_\phi) > \delta_{\text{PROPS}}$ **then**
10:            **return** $\boldsymbol{\phi}$
11: **return** $\boldsymbol{\phi}$

---

typically parameterized as a Gaussian $\mathcal{N}(\boldsymbol{\mu}(\boldsymbol{s}), \Sigma(\boldsymbol{s}))$ with mean $\boldsymbol{\mu}(\boldsymbol{s})$ and diagonal covariance matrix $\Sigma(\boldsymbol{s})$. Since actions in the tail of the Gaussian far from the mean will always be under-sampled, the ROS update will continually push the components of $\boldsymbol{\mu}(\boldsymbol{s})$ towards $\pm\infty$ and the diagonal components of $\Sigma(\boldsymbol{s})$ towards 0 to increase the probability of sampling these actions. The result is a degenerate behavior policy that is so far from the target policy that sampling from it increases sampling error.[3] We illustrate this scenario with 1-dimensional continuous actions in Fig. 6 of Appendix B.

To address these challenges, we propose a new behavior policy update. To address Challenge 1, first observe that the gradient of the ROS loss $\nabla_\phi \mathcal{L} = \nabla_\phi \log \pi_\phi(\boldsymbol{a}|\boldsymbol{s})|_{\phi=\theta}$ is equivalent to the policy gradient (Eq. 2) with $A^{\pi_\theta}(\boldsymbol{s}, \boldsymbol{a}) = -1, \forall(\boldsymbol{s}, \boldsymbol{a})$. Since the clipped surrogate objective of PPO (Eq. 3) prevents destructively large updates in on-policy policy gradient learning, we use a similar clipped surrogate objective in place of the ROS objective:

$$\mathcal{L}_{\text{CLIP}}(\boldsymbol{s}, \boldsymbol{a}, \boldsymbol{\phi}, \boldsymbol{\theta}, \epsilon_{\text{PROPS}}) = \min\left[ -\frac{\pi_\phi(\boldsymbol{a}|\boldsymbol{s})}{\pi_\theta(\boldsymbol{a}|\boldsymbol{s})}, -\texttt{clip}\left( \frac{\pi_\phi(\boldsymbol{a}|\boldsymbol{s})}{\pi_\theta(\boldsymbol{a}|\boldsymbol{s})}, 1 - \epsilon_{\text{PROPS}}, 1 + \epsilon_{\text{PROPS}} \right) \right]. \quad (5)$$

Table 1 in Appendix B summarizes the behavior of $\mathcal{L}_{\text{CLIP}}$. Intuitively, this objective is equivalent to the PPO objective (Eq. 3) with $A(\boldsymbol{s}, \boldsymbol{a}) = -1, \forall(\boldsymbol{s}, \boldsymbol{a})$ and incentivizes the agent to decrease the probability of observed actions by at most a factor of $1 - \epsilon_{\text{PROPS}}$. Let $g(\boldsymbol{s}, \boldsymbol{a}, \boldsymbol{\phi}, \boldsymbol{\theta}) = \frac{\pi_\phi(\boldsymbol{a}|\boldsymbol{s})}{\pi_\theta(\boldsymbol{a}|\boldsymbol{s})}$. When $g(\boldsymbol{s}, \boldsymbol{a}, \boldsymbol{\phi}, \boldsymbol{\theta}) < 1 - \epsilon_{\text{PROPS}}$, this objective is clipped at $-(1 - \epsilon_{\text{PROPS}})$. The loss gradient $\nabla_\phi \mathcal{L}_{\text{CLIP}}$ becomes zero, and the $(\boldsymbol{s}, \boldsymbol{a})$ pair has no effect on the policy update. When $g(\boldsymbol{s}, \boldsymbol{a}, \boldsymbol{\phi}, \boldsymbol{\theta}) > 1 - \epsilon_{\text{PROPS}}$, clipping does not apply, and the gradient $\nabla_\phi \mathcal{L}_{\text{CLIP}}$ points in a direction that decreases the probability of $\pi_\phi(\boldsymbol{a}|\boldsymbol{s})$. As in the PPO update, this clipping mechanism avoids destructively large policy updates and permits us to perform many epochs of minibatch updates with the same batch of data.

To address the second challenge and prevent degenerate behavior policies, we introduce an auxiliary loss that incentivizes the agent to minimize the KL divergence between the behavior policy and target policy at states in the observed data. The full PROPS objective is then:

$$\mathcal{L}_{\text{PROPS}}(\boldsymbol{s}, \boldsymbol{a}, \boldsymbol{\phi}, \boldsymbol{\theta}, \epsilon_{\text{PROPS}}, \lambda) = \mathcal{L}_{\text{CLIP}}(\boldsymbol{s}, \boldsymbol{a}, \boldsymbol{\phi}, \boldsymbol{\theta}) - \lambda D_{\text{KL}}(\pi_\theta(\cdot|\boldsymbol{s}) || \pi_\phi(\cdot|\boldsymbol{s})) \quad (6)$$

where $\lambda$ is a regularization coefficient quantifying a trade-off between maximizing $\mathcal{L}_{\text{PROPS}}$ and minimizing $D_{\text{KL}}$. We provide full pseudocode for the PROPS update in Algorithm 2. Like ROS, we set the behavior policy parameters $\boldsymbol{\phi}$ equal to the target policy parameters at the start of each behavior update, and then make a local adjustment to $\boldsymbol{\phi}$ to increase the probabilities of under-sampled actions.

---

[3]This challenge is specific to continuous-action tasks and does not arise in discrete-action tasks.

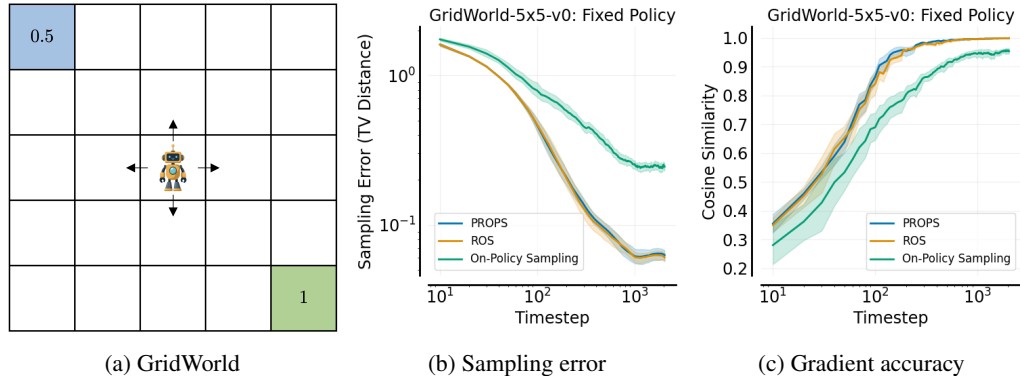

(a) GridWorld        (b) Sampling error        (c) Gradient accuracy

Figure 2: **(a)** A GridWorld task in which the agent receives reward $+1$ upon reaching the bottom right corner (the optimal goal), a reward of $+0.5$ upon reaching the top left corner (the suboptimal goal), and a reward of $-0.01$. The agent always starts in the center of the grid. Under an initially uniform policy, the agent visits both goals with equal probability, and thus the true policy gradient increases the probability of reaching the optimal goal. **(b, c)** PROPS reduces sampling error and achieves more accurate gradients faster than on-policy sampling.

We stop the PROPS update when $D_{\mathrm{KL}}(\pi_{\boldsymbol{\theta}}||\pi_{\boldsymbol{\phi}})$ reaches a chosen threshold $\delta_{\mathrm{PROPS}}$. This technique further safeguards against large policy updates and is used in widely adopted implementations of PPO (Raffin et al., 2021; Liang et al., 2018). The PROPS update allows us to efficiently learn a behavior policy that keeps the distribution of data in the buffer close to the expected distribution of the target policy.

## 6 EXPERIMENTS

The central goal of our work is to understand whether on-policy policy gradient algorithms are more data efficient learners when they use on-policy data acquired *without* on-policy sampling. Towards this goal, we design experiments to answer the two questions:

**Q1:** Does PROPS achieve lower sampling error than on-policy sampling during training?

**Q2:** Does PROPS increase the fraction of training runs that converge to high-return policies and improve the data efficiency of on-policy policy gradient algorithms?

Our empirical analysis focuses on continuous-state continuous-action MuJoCo benchmark tasks and a tabular 5x5 GridWorld task (Fig. 2a). We additionally consider three continuous-state discrete-action tasks: CartPole-v1, LunarLander-v2, and Discrete2D100-v0 – a 2D navigation task with 100 discrete actions. Due to space constraints, we include these tasks in Appendix D.4.

### 6.1 CORRECTING SAMPLING ERROR FOR A FIXED TARGET POLICY

We first study how quickly PROPS decreases sampling error when the target policy is fixed. This setting is similar to the policy evaluation setting considered by Zhong et al. (2022). As such, we provide two baselines for comparison: on-policy sampling and ROS.

**Sampling error metrics**. In GridWorld, we compute sampling error as the total variation (TV) distance between the empirical state-action visitation $d_{\mathcal{D}}(\boldsymbol{s}, \boldsymbol{a})$ distribution – denoting the proportion of times $(\boldsymbol{s}, \boldsymbol{a})$ appears in buffer $\mathcal{D}$ – and the true state-action visitation distribution under the agent's policy: $\sum_{(\boldsymbol{s},\boldsymbol{a})\in\mathcal{D}} |d_{\mathcal{D}}(\boldsymbol{s}, \boldsymbol{a}) - d_{\pi_{\boldsymbol{\theta}}}(\boldsymbol{s}, \boldsymbol{a})|$. In continuous MuJoCo tasks where it is difficult to compute $d_{\mathcal{D}}(\boldsymbol{s}, \boldsymbol{a})$, we follow Zhong et al. (2022) and measure sampling error using the KL-divergence $D_{\mathrm{KL}}(\pi_{\mathcal{D}}||\pi_{\boldsymbol{\theta}})$ between the empirical policy $\pi_{\mathcal{D}}$ and the target policy $\pi_{\boldsymbol{\theta}}$. We estimate $\pi_{\mathcal{D}}$ as the maximum likelihood estimate under data in the buffer via stochastic gradient ascent. Further details on how we compute $\pi_{\mathcal{D}}$ are in Appendix C.

Since it is straightforward to compute the true policy gradient in the GridWorld task, we additionally investigate how sampling error reduction affects gradient estimation by measuring the cosine simi-

larity between the empirical policy gradient $\nabla_{\boldsymbol{\theta}} \widehat{J}(\boldsymbol{\theta})$ and the true policy gradient. As the empirical gradient aligns more closely with the true gradient, the cosine similarity approaches 1.

**Experimental setup**. In all tasks, we use a buffer with capacity of $\lfloor T/2 \rfloor$ samples, where $T$ is the total number of samples collected by the agent. Thus, we expect sampling error to decrease over the first $\lfloor T/2 \rfloor$ samples and then remain roughly constant afterwards once the buffer is full. We use randomly initialized target policies. Further experimental details such as hyperparameter tuning are described in Appendix D.1.

**Results.** As shown in Fig. 2b and 2c, in Grid-World, PROPS decreases sampling error faster than on-policy sampling, resulting in more accurate policy gradient estimates. PROPS and ROS to perform similarly, though this behavior is expected; in a tabular setting with a fixed target policy (*i.e.* there is no off-policy data in the buffer), we do not encounter Challenge 1 and 2 described in the previous section. In continu-

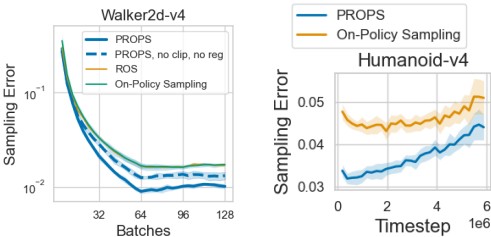

(a) Fixed target policy.   (b) During RL training.

Figure 3: PROPS reduces sampling error faster than on-policy sampling and ROS. In (a), the ROS and on-policy sampling curves overlap. Solid curves denote means over 5 seeds. Shaded regions denote 95% confidence intervals.

ous MuJoCo tasks where Challenge 2 arises, PROPS decreases sampling error faster than on-policy sampling and ROS (Fig. 3a). In fact, ROS shows little to no improvement over on-policy sampling in every MuJoCo task. This limitation of ROS is unsurprising, as Zhong et al. (2022) showed that ROS struggled to reduce sampling error even in low-dimensional continuous-action tasks. Moreover, PROPS decreases sampling error without clipping and regularization, emphasizing how adaptive off-policy sampling alone decreases sampling error. Due to space constraints, we include results for the remaining environments in Appendix D.1. We additionally include experiments using a fixed, randomly initialized target policy as well as ablation studies isolating the effects of PROPS's objective clipping and regularization in Appendix D.1. Results with a random target policy are qualitatively similar to those in Fig. 3a, and we observe that clipping and regularization both individually help reduce sampling error.

## 6.2 CORRECTING SAMPLING ERROR DURING RL TRAINING

We are ultimately interested in understanding how replacing on-policy sampling with PROPS affects the data efficiency of on-policy learning, where the target policy is continually updated. In the following experiments, we train RL agents with PROPS and on-policy sampling to evaluate (1) the data efficiency of training, (2) the distribution of returns achieved at the end of training, and (3) the sampling error throughout training. We use the same sampling error metrics described in the previous section and measure data efficiency as the return achieved within a fixed training budget. Since ROS (Zhong et al., 2022) is computationally expensive and fails to reduce sampling error in MuJoCo tasks even with a fixed policy, we omit it from MuJoCo experiments.

**Experimental setup.** We use PPO (Schulman et al., 2017) to update the target policy. We consider two baseline methods for providing data to compute PPO updates: (1) vanilla PPO with on-policy sampling, and (2) PPO with on-policy sampling and a buffer of size $b$ (PPO-BUFFER). PPO-BUFFER is a naive method for improving data efficiency of on-policy algorithms by reusing off-policy data collected by old target policies as if it were on-policy data. Although PPO-BUFFER computes biased gradients, it has been successfully applied in difficult learning tasks (Berner et al., 2019). Since PROPS and PPO-BUFFER have access to the same amount of data for each policy update, any performance difference between these two methods can be attributed to differences in how they sample actions during data collection.

In MuJoCo experiments, we set $b = 2$ such that agents retain a batch of data for one extra iteration before discarding it. In GridWorld, we use $b = 1$ and discard all historic data. Since PROPS and PPO-BUFFER compute target policy updates with $b$ times as much learning data as PPO, we integrate this extra data by increasing the minibatch size for target and behavior policy updates by a factor

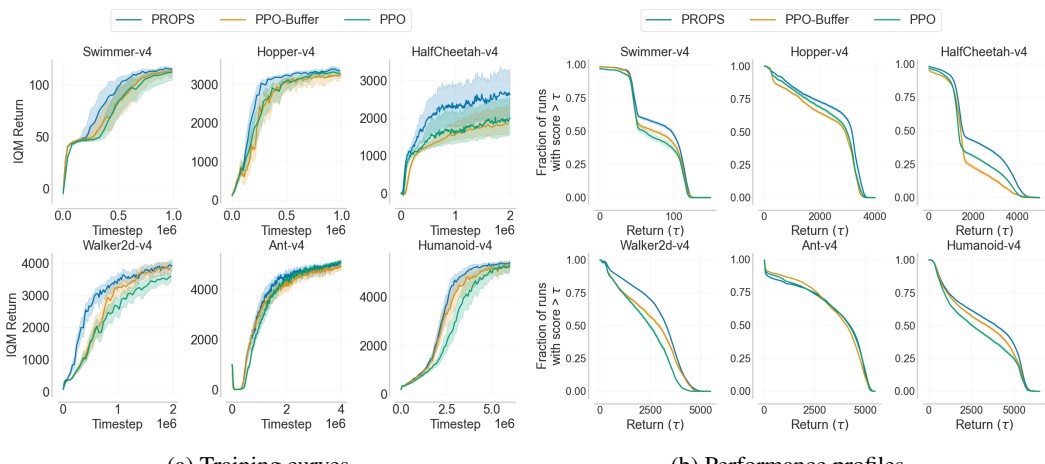

(a) Training curves.                    (b) Performance profiles.

Figure 4: **(a)** IQM returns over 50 seeds. Shaded regions denote 95% bootstrap confidence intervals. **(b)** Performance profiles over 50 seeds. Higher values correspond to more reliable convergence to high-return policies. Shaded regions denote 95% bootstrap confidence intervals.

of $b$. Further experimental details including hyperparameter tuning are described in Appendix E. For MuJoCo tasks, we plot the interquartile mean (IQM) return throughout training as well as the distribution of returns achieved at the end of training (*i.e.*, the performance profile) (Agarwal et al., 2021). For GridWorld, we plot the agent's success rate, the fraction of times it finds the optimal goal.

**Results.** As shown in Fig. 5a, on-policy sampling has approximately a 77% success rate on Grid-World, whereas PROPS and ROS achieve 100% success rate. In Fig. 4a, PROPS achieves higher return than both PPO and PPO-BUFFER throughout training in all MuJoCo tasks except Ant-v4 where PROPS dips slightly below PPO's return near the end of training. Moreover, in Fig. 4b, the performance profile of PROPS almost always lies above the performance profiles of PPO and PPO-BUFFER, indicating that any given run of PROPS is more like to obtain a higher return than PPO-BUFFER. Thus, we affirmatively answer **Q2** posed at the start of this section: PROPS increases the fraction of training runs with high return and increases data efficiency.

In Appendix D.4, we provide additional experiments demonstrating that PROPS improves data efficiency in discrete-action tasks. We additionally ablate the buffer size $b$ in Appendix D.2. We find that data efficiency may decrease with a larger buffer size. Intuitively, the more historic data kept around, the more data that must be collected to impact the aggregate data distribution.

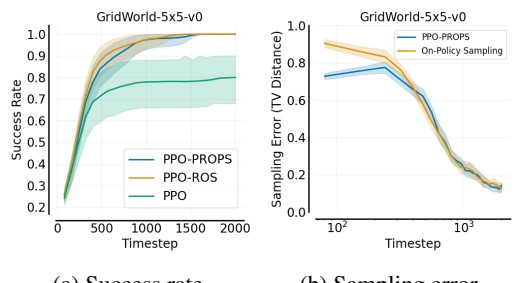

(a) Success rate.          (b) Sampling error.

Figure 5: GridWorld RL experiments over 50 seeds.

Having established that PROPS improves data efficiency, we now investigate if PROPS is appropriately adjusting the data distribution of the buffer by comparing the sampling error achieved throughout training with PROPS and PPO-BUFFER. Training with PROPS produces a different sequence of target policies than training with PPO-BUFFER produces. To provide a fair comparison, we compute sampling error for PPO-BUFFER using the target policy sequence produced by PROPS. More concretely, we fill a second buffer with on-policy samples collected by the *target policies* produced while training with PROPS and then compute the sampling error using data in this buffer.

As shown in Fig. 5b, PROPS achieves lower sampling error than on-policy sampling with a buffer in Humanoid-v4. Due to space constraints, we provide sampling error curves for the remaining MuJoCo environments in Appendix D.2. In GridWorld, PROPS and ROS reduce sampling error in

the first 300 steps and closely matches on-policy sampling afterwards. We use a batch size of 80 in these experiments, and as the target policy becomes more deterministic, larger batch sizes are needed to observe differences between PROPS and on-policy sampling.[4]

We additionally ablate the effects of the clipping coefficient $\epsilon_{\text{PROPS}}$ and regularization coefficient $\lambda$ in Appendix D.2. Without clipping or without regularization, PROPS often achieves greater sampling error than on-policy sampling, indicating that both help to keep sampling error low. Moreover, data efficiency generally decreases when we remove clipping or regularization, showing both are essential to PROPS. Thus, we affirmatively answer **Q1** posed at the start of this section: PROPS achieves lower sampling error than on-policy sampling when the target policy is fixed and during RL training.

## 7 DISCUSSION

This work has shown that adaptive, off-policy sampling can be used to reduce sampling error in data collected throughout RL training and improve the data efficiency of on-policy policy gradient algorithms. We have introduced an algorithm that scales adaptive off-policy sampling to continuous control RL benchmarks and enables tracking of the changing on-policy distribution. By integrating this data collection procedure into the popular PPO algorithm, the main conclusion of our analysis is that on-policy learning algorithms learn most efficiently with on-policy data, *not* on-policy sampling. In this section, we discuss limitations of our work and present opportunities for future research.

PROPS builds upon the ROS algorithm of Zhong et al. (2022). While Zhong et al. (2022) focused on theoretical analysis and policy evaluation in small scale domains, we chose to focus on empirical analysis with policy learning in standard RL benchmarks. An important direction for future work would be theoretical analysis of PROPS, in particular whether PROPS also enjoys the same faster convergence rate that was shown for ROS relative to on-policy sampling.

A limitation of PROPS is that the update indiscriminately increases the probability of under-sampled actions without considering their importance in gradient computation. For instance, if an under-sampled action has zero advantage, it has no impact on the gradient and need not be sampled. An interesting direction for future work could be to prioritize correcting sampling error for $(s, a)$ that have the largest influence on the gradient estimate, *i.e.*, large advantage (positive or negative).

Beyond these more immediate directions, our work opens up other opportunities for future research. A less obvious feature of the PROPS behavior policy update is that it can be used track the empirical data distribution of *any* desired policy, not only that of the current policy. This feature means PROPS has the potential to be integrated into off-policy RL algorithms and used so that the empirical distribution more closely matches a desired exploration distribution.f Thus, PROPS could be used to perform focused exploration without explicitly tracking state and action counts.

## 8 CONCLUSION

In this paper, we ask whether on-policy policy gradient methods are more data efficient using on-policy sampling or on-policy *data* acquired *without* on-policy sampling. To answer this question, we introduce an adaptive, *off-policy* sampling method for on-policy policy gradient learning that collects data such that the empirical distribution of sampled actions closely matches the expected on-policy data distribution at observed states. Our method, Proximal Robust On-policy Sampling (PROPS), periodically updates the data collecting behavior policy so as to increase the probability of sampling actions that are currently under-sampled with respect to the on-policy distribution. Furthermore, rather than discarding collected data after every policy update, PROPS permits more data efficient on-policy learning by using data collection to adjust the distribution of previously collected data to be approximately on-policy. We replace on-policy sampling with PROPS to generate data for the widely-used PPO algorithm and empirically demonstrate that PROPS produces data that more closely matches the expected on-policy distribution and yields more data efficient learning compared to on-policy sampling.

---

[4]When the target policy is deterministic, we always have zero sampling error, and PROPS will exactly match on-policy sampling.

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

# Appendix

## Table of Contents

## A  THEORETICAL RESULTS

In this section, we present the proof of Proposition 2. We use $d_m$, $\pi_m$, and $p_m$ as the empirical state visitation distribution, empirical policy, and empirical transition probabilities after $m$ state-action pairs have been taken, respectively. That is, $d_m(s)$ is the proportion of the $m$ states that are $s$, $\pi_m(a|s)$ is the proportion of the time that action $a$ was observed in state $s$, and $p_m(s'|s,a)$ is the proportion of the time that the state changed to $s'$ after action $a$ was taken in state $s$.

**Proposition 2.** *Assume that data is collected with an adaptive behavior policy that always takes the most under-sampled action in each state, $s$, with respect to policy $\pi$, i.e, $a \leftarrow \arg\max_{a'}(\pi(a'|s) - \pi_m(a'|s))$. We further assume that $\mathcal{S}$ and $\mathcal{A}$ are finite. Then we have that the empirical state visitation distribution, $d_m$, converges to the state distribution of $\pi$, $d_\pi$, with probability $1$:*

$$\forall s, \lim_{m\to\infty} d_m(s) = d_\pi(s).$$

*Proof.* The proof of this theorem builds upon Lemma 1 and 2 by Zhong et al. (2022). Note that these lemmas superficially concern the ROS method whereas we are interested in data collection by taking the most under-sampled action at each step. However, as stated in the proof by Zhong et al. (2022), these methods are equivalent under an assumption they make about the step-size parameter of the ROS method. Thus, we can immediately adopt these lemmas for this proof.

Under Lemma 1 of Zhong et al. (2022), we have that $\lim_{m\to\infty} \pi_m(a|s) = \pi(a|s)$ for any state $s$ under this adaptive data collection procedure. We then have the following $\forall s$:

$$\lim_{m\to\infty} d_m(s) \stackrel{(a)}{=} \lim_{m\to\infty} \sum_{\tilde{s}} \sum_{\tilde{a}} p_m(s|\tilde{s},\tilde{a}) \pi_m(\tilde{a}|\tilde{s}) d_m(\tilde{s})$$

$$= \sum_{\tilde{s}} \sum_{\tilde{a}} \lim_{m\to\infty} p_m(s|\tilde{s},\tilde{a}) \pi_m(\tilde{a}|\tilde{s}) d_m(\tilde{s})$$

$$= \sum_{\tilde{s}} \sum_{\tilde{a}} \lim_{m\to\infty} p_m(s|\tilde{s},\tilde{a}) \lim_{m\to\infty} \pi_m(\tilde{a}|\tilde{s}) \lim_{m\to\infty} d_m(\tilde{s})$$

$$\stackrel{(b)}{=} \sum_{\tilde{s}} \sum_{\tilde{a}} p(s|\tilde{s},\tilde{a}) \pi(\tilde{a}|\tilde{s}) \lim_{m\to\infty} d_m(\tilde{s}).$$

Here, (a) follows from the fact that the empirical frequency of state $s$ can be obtained by considering all possible transitions that lead to $s$. The last line, (b), holds with probability 1 by the strong law of large numbers and Lemma 2 of Zhong et al. (2022).

We now have a system of $|\mathcal{S}|$ variables and $|\mathcal{S}|$ linear equations. Define variables $x(s) := \lim_{m\to\infty} d_m(s)$ and let $\boldsymbol{x} \in \mathbf{R}^{|\mathcal{S}|}$ be the vector of these variables. We then have $x = P^\pi x$ where $P^\pi \in \mathbf{R}^{|\mathcal{S}| \times |\mathcal{S}|}$ is the transition matrix of the Markov chain induced by running policy $\pi$. Assuming that this Markov chain is irreducible, $d_\pi$ is the unique solution to this system of equations and hence $\lim_{m\to\infty} d_m(s) = d_\pi(s), \forall s$.

$\square$

Next, we provide additional theory to describe the relationship between different hyperparameters in PROPS:

1. The amount of sampling error in previously collected data and the size of behavior policy updates.

2. The amount of historic data retained by an agent and the amount of additional data the behavior policy must collect to reduce sampling error.

For simplicity, we first focus on a simple bandit setting and then extend to a tabular RL setting.

Suppose we have already collected $m$ state-action pairs and these have been observed with empirical distribution $\pi_m(\boldsymbol{a})$. From what distribution should we sample an additional $k$ state-action pairs so that the empirical distribution over the $m + k$ samples is equal in expectation to $\pi_{\boldsymbol{\theta}}$?

**Proposition 3.** *Assume that $m$ actions have been collected by running some policy $\pi_{\boldsymbol{\theta}}(\boldsymbol{a})$ and $\pi_m(\boldsymbol{a})$ is the empirical distribution on this dataset. If we collect an additional $k$ state-action pairs using the following distribution, and if $(m + k)\pi_{\boldsymbol{\theta}}(\boldsymbol{a}) \geq m \cdot \pi_m(\boldsymbol{a})$, then the aggregate empirical distribution over the $m + k$ pairs is equal to $\pi_{\boldsymbol{\theta}}(\boldsymbol{a})$ in expectation:*

$$\pi_b(\boldsymbol{a}) := \frac{1}{Z}\left[\pi_{\boldsymbol{\theta}}(\boldsymbol{a}) + \frac{m}{k}\left(\pi_{\boldsymbol{\theta}}(\boldsymbol{a}) - \pi_m(\boldsymbol{a})\right)\right]$$

*where $Z = \sum_{\boldsymbol{a}\in\mathcal{A}}\left[\pi_{\boldsymbol{\theta}}(\boldsymbol{a}) + \frac{m}{k}\left(\pi_{\boldsymbol{\theta}}(\boldsymbol{a}) - \pi_m(\boldsymbol{a})\right)\right]$ is a normalization coefficient.*

*Proof.* Observe that $(m + k)\pi_{\boldsymbol{\theta}}(\boldsymbol{a})$ is the expected number of times $\boldsymbol{a}$ is sampled under $\pi_{\boldsymbol{\theta}}$ after $m + k$ steps, $m \cdot \pi_m(\boldsymbol{a})$ is the number of times each $\boldsymbol{a}$ was sampled thus far, and $k \cdot \pi_b(\boldsymbol{a})$ is the expected number of times $\boldsymbol{a}$ is sampled under our behavior policy after $k$ steps. We want to choose $\pi_b(\boldsymbol{a})$ such that $(m + k)\pi_{\boldsymbol{\theta}}(\boldsymbol{a}) = m \cdot \pi_m(\boldsymbol{a}) + k \cdot \pi_b(\boldsymbol{a})$ in expectation.

$$(m + k)\pi_{\boldsymbol{\theta}}(\boldsymbol{a}) = k \cdot \pi_b(\boldsymbol{a}) + m \cdot \pi_m(\boldsymbol{a})$$
$$-k \cdot \pi_b(\boldsymbol{a}) = m \cdot \pi_m(\boldsymbol{a}) - (m + k)\pi_{\boldsymbol{\theta}}(\boldsymbol{a})$$
$$\pi_b(\boldsymbol{a}) = -\frac{m}{k}\pi_m(\boldsymbol{a}) + \left(\frac{m}{k} + 1\right)\pi_{\boldsymbol{\theta}}(\boldsymbol{a})$$
$$= \pi_{\boldsymbol{\theta}}(\boldsymbol{a}) + \frac{m}{k}\left(\pi_{\boldsymbol{\theta}}(\boldsymbol{a}) - \pi_m(\boldsymbol{a})\right)$$

Note that $\pi_b(\boldsymbol{a})$ will be a valid probability distribution after normalizing only if

$$\pi_{\boldsymbol{\theta}}(\boldsymbol{a}) + \frac{m}{k}\left(\pi_{\boldsymbol{\theta}}(\boldsymbol{a}) - \pi_m(\boldsymbol{a})\right) \geq 0$$
$$\left(\frac{m}{k} + 1\right)\pi_{\boldsymbol{\theta}}(\boldsymbol{a}) \geq \frac{m}{k}\pi_m(\boldsymbol{a})$$
$$(m + k)\pi_{\boldsymbol{\theta}}(\boldsymbol{a}) \geq m \cdot \pi_m(\boldsymbol{a}).$$

If $(m + k)\pi_{\boldsymbol{\theta}}(\boldsymbol{a}) < m \cdot \pi_m(\boldsymbol{a})$, then prior to collecting additional data with our behavior policy, $\boldsymbol{a}$ already appears in our data more times in our data than it would in expectation after $m + k$ steps under $\pi_{\boldsymbol{\theta}}$. In other words, we would need to collect more than $k$ additional samples to achieve zero sampling error (or discard some previously collected samples).

$\square$

**When sampling error is large, behavior policy updates must also be large.** Intuitively, the difference $\pi_{\boldsymbol{\theta}}(\boldsymbol{a}) - \pi_m(\boldsymbol{a})$ is the mismatch between the true and empirical visitation distributions, so adding this term to $d_{\pi_{\boldsymbol{\theta}}}$ adjusts $d_{\pi_{\boldsymbol{\theta}}}$ to reduce this mismatch. If $\pi_{\boldsymbol{\theta}}(\boldsymbol{a}) - \pi_m(\boldsymbol{a}) < 0$, then $\boldsymbol{a}$ is over-sampled w.r.t $\pi_{\boldsymbol{\theta}}$, and $\pi_b$ will decrease the probability of sampling $\boldsymbol{a}$. If $\pi_{\boldsymbol{\theta}}(\boldsymbol{a}) - \pi_m(\boldsymbol{a}) > 0$, then $\boldsymbol{a}$ is under-sampled w.r.t $\pi_{\boldsymbol{\theta}}$, and $\pi_b$ will increase the probability of sampling $\boldsymbol{a}$. When $|\pi_{\boldsymbol{\theta}}(\boldsymbol{a}) - \pi_m(\boldsymbol{a})|$ is small, the optimal $\pi_b(\boldsymbol{a})$ requires only a small adjustment from $\pi_{\boldsymbol{\theta}}$ (*i.e.*, a small update to the behavior policy is sufficient to reduce sampling error). When $|\pi_{\boldsymbol{\theta}}(\boldsymbol{a}) - \pi_m(\boldsymbol{a})|$ is large, the optimal $\pi_b(\boldsymbol{a})$ requires a large adjustment from $\pi_{\boldsymbol{\theta}}$ (*i.e.*, a large to the behavior policy is needed to reduce sampling error).

**When we retain large amounts of historic data, the behavior policy must collect a large amount of additional data to reduce sampling error in the aggregate distribution.** The $\frac{m}{k}$ factor implies that how much we adjust $d_{\pi_{\boldsymbol{\theta}}}$ depends on how much data we have already collected ($m$) and how much additional data we will collect ($k$). If the $k$ additional samples to collect represent a small fraction of the aggregate $m + k$ samples (*i.e.* $k << m$), then $\frac{m}{k}$ is large, and the adjustment to $d_{\pi_{\boldsymbol{\theta}}}$ is large. This case generally arises when we retain more and more historic data. If the $k$ additional samples to collect represent a large fraction of the aggregate $m + k$ samples (*i.e.* $k >> m$), then $\frac{m}{k}$ is small, and the adjustment to $d_{\pi_{\boldsymbol{\theta}}}$ is small. This case generally arises when we retain little to no historic data.

The next proposition extends this analysis to the tabular RL setting.

**Proposition 4.** *Assume that $m$ state-action pairs have been collected by running some policy and $d_m(\boldsymbol{s}, \boldsymbol{a})$ is the empirical distribution on this dataset. If we collect an additional $k$ state-action pairs using the following distribution, and if $(m + k)d_{\pi_{\boldsymbol{\theta}}}(\boldsymbol{s}, \boldsymbol{a}) \geq m \cdot d_m(\boldsymbol{s}, \boldsymbol{a})$, then the aggregate empirical distribution over the $m + k$ pairs is equal to $d_{\pi_{\boldsymbol{\theta}}}(\boldsymbol{s}, \boldsymbol{a})$ in expectation:*

$$d_b(\boldsymbol{s}, \boldsymbol{a}) := \frac{1}{Z}\left[d_{\pi_{\boldsymbol{\theta}}}(\boldsymbol{s}, \boldsymbol{a}) + \frac{m}{k}\left(d_{\pi_{\boldsymbol{\theta}}}(\boldsymbol{s}, \boldsymbol{a}) - d_m(\boldsymbol{s}, \boldsymbol{a})\right)\right]$$

*where $Z = \sum_{(\boldsymbol{s}, \boldsymbol{a}) \in \mathcal{S} \times \mathcal{A}}\left[d_{\pi_{\boldsymbol{\theta}}}(\boldsymbol{s}, \boldsymbol{a}) + \frac{m}{k}\left(d_{\pi_{\boldsymbol{\theta}}}(\boldsymbol{s}, \boldsymbol{a}) - d_m(\boldsymbol{s}, \boldsymbol{a})\right)\right]$ is a normalization coefficient.*

*Proof.* The proof is identical to the proof of Proposition 3, replacing $\pi_{\boldsymbol{\theta}}(\boldsymbol{a}), \pi_m(\boldsymbol{a})$, and $\pi_b(\boldsymbol{a})$ with $d_{\pi_{\boldsymbol{\theta}}}(\boldsymbol{s}, \boldsymbol{a}), d_m(\boldsymbol{s}, \boldsymbol{a})$, and $d_b(\boldsymbol{s}, \boldsymbol{a})$. $\qquad\square$

In practice, we cannot sample directly from the visitation distribution $d_b(\boldsymbol{s}, \boldsymbol{a})$ in Proposition 4 and instead approximate sampling from this distribution by sampling from its corresponding policy $\pi_b(\boldsymbol{a}|\boldsymbol{s}) = d_b(\boldsymbol{s}, \boldsymbol{a}) / \sum_{(\boldsymbol{s}', \boldsymbol{a}') \in \mathcal{S} \times \mathcal{A}} d_b(\boldsymbol{s}', \boldsymbol{a}')$.

# B  PROPS IMPLEMENTATION DETAILS

In this appendix, we describe two relevant implementation details for the PROPS update (Algorithm 2). We additionally summarize the behavior of PROPS's clipping mechanism in Table 1.

1. **PROPS update:** The PROPS update adapts the behavior policy to reduce sampling error in the buffer $\mathcal{D}$. When performing this update with a full buffer, we exclude the oldest batch of data collected by the behavior policy (*i.e.*, the $m$ oldest transitions in $\mathcal{D}$); this data will be evicted from the buffer before the next behavior policy update and thus does not contribute to sampling error in $\mathcal{D}$.

2. **Behavior policy class:** We compute behavior policies from the same policy class used for target policies. In particular, we consider Gaussian policies which output a mean $\mu(\boldsymbol{s})$ and a variance $\sigma^2(\boldsymbol{s})$ and then sample actions $\boldsymbol{a} \sim \pi(\cdot|\boldsymbol{s}) \equiv \mathcal{N}(\mu(\boldsymbol{s}), \sigma^2(\boldsymbol{s}))$. In principle, the target and behavior policy classes can be different. However, using the same class for both policies allows us to easily initialize the behavior policy equal to the target policy at the start of each update. This initialization is necessary to ensure the PROPS update increases the probability of sampling actions that are currently under-sampled with respect to the target policy.

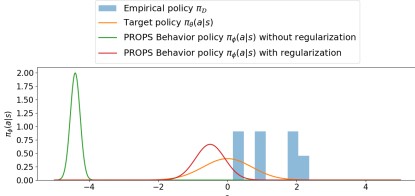

Figure 6: In this example, $\pi(\cdot|\boldsymbol{s}) = \mathcal{N}(0, 1)$. After several visits to $\boldsymbol{s}$, all sampled actions (blue) satisfy $a > 0$ so that actions $a < 0$ are under-sampled. Without regularization, PROPS will attempt to increase the probabilities of under-sampled action in the tail of target policy distribution (green). The regularization term in the PROPS objective ensures the behavior policy remains close to target policy.

| $g(\boldsymbol{s}\,\boldsymbol{a}, \boldsymbol{\phi}, \boldsymbol{\theta}) > 0$ | Is the objective clipped? | Return value of min | Gradient |
|---|---|---|---|
| $g(\boldsymbol{s}\,\boldsymbol{a}, \boldsymbol{\phi}, \boldsymbol{\theta}) \in [1 - \epsilon_{\text{PROPS}}, 1 + \epsilon_{\text{PROPS}}]$ | No | $-g(\boldsymbol{s}, \boldsymbol{a}, \boldsymbol{\phi}, \boldsymbol{\theta})$ | $\nabla_{\boldsymbol{\phi}} \mathcal{L}_{\text{CLIP}}$ |
| $g(\boldsymbol{s}, \boldsymbol{a}, \boldsymbol{\phi}, \boldsymbol{\theta}) > 1 + \epsilon_{\text{PROPS}}$ | No | $-g(\boldsymbol{s}, \boldsymbol{a}, \boldsymbol{\phi}, \boldsymbol{\theta})$ | $\nabla_{\boldsymbol{\phi}} \mathcal{L}_{\text{CLIP}}$ |
| $g(\boldsymbol{s}, \boldsymbol{a}, \boldsymbol{\phi}, \boldsymbol{\theta}) < 1 - \epsilon_{\text{PROPS}}$ | Yes | $-(1 - \epsilon_{\text{PROPS}})$ | $\mathbf{0}$ |

Table 1: Behavior of PROPS's clipped surrogate objective (Eq. 5).

## C  COMPUTING SAMPLING ERROR

We claim that PROPS improves the data efficiency of on-policy learning by reducing sampling error in the agent's buffer $\mathcal{D}$ with respect to the agent's current (target) policy. To measure sampling error, we use the KL-divergence $D_{\text{KL}}(\pi_{\mathcal{D}}||\pi_{\boldsymbol{\theta}})$ between the empirical policy $\pi_{\mathcal{D}}$ and the target policy $\pi_{\boldsymbol{\theta}}$ which is the primary metric Zhong et al. (2022) used to show ROS reduces sampling error:

$$D_{\text{KL}}(\pi_{\mathcal{D}}||\pi_{\boldsymbol{\theta}}) = \mathbb{E}_{\boldsymbol{s}\sim\mathcal{D}, \boldsymbol{a}\sim\pi_{\mathcal{D}}(\cdot|\boldsymbol{s})} \left[ \log \left( \frac{\pi_{\mathcal{D}}(\boldsymbol{a}|\boldsymbol{s})}{\pi_{\boldsymbol{\theta}}(\boldsymbol{a}|\boldsymbol{s})} \right) \right]. \tag{7}$$

We compute a parametric estimate of $\pi_{\mathcal{D}}$ by maximizing the log-likelihood of $\mathcal{D}$ over the same policy class used for $\pi_{\boldsymbol{\theta}}$. More concretely, we let $\boldsymbol{\theta}'$ be the parameters of neural network with the same architecture as $\pi_{\boldsymbol{\theta}}$ train and then compute:

$$\boldsymbol{\theta}_{\text{MLE}} = \arg\max_{\boldsymbol{\theta}'} \sum_{(\boldsymbol{s}, \boldsymbol{a}) \in \mathcal{D}} \log \pi_{\boldsymbol{\theta}'}(\boldsymbol{a}|\boldsymbol{s}) \tag{8}$$

using stochastic gradient ascent. After computing $\boldsymbol{\theta}_{\text{MLE}}$, we then estimate sampling error using the Monte Carlo estimator:

$$D_{\text{KL}}(\pi_{\mathcal{D}}||\pi_{\boldsymbol{\theta}}) \approx \sum_{(\boldsymbol{s}, \boldsymbol{a}) \in \mathcal{D}} \left( \log \pi_{\boldsymbol{\theta}_{\text{MLE}}}(\boldsymbol{a}|\boldsymbol{s}) - \log \pi_{\boldsymbol{\theta}}(\boldsymbol{a}|\boldsymbol{s}) \right). \tag{9}$$

## D  ADDITIONAL EXPERIMENTS

In this appendix, we include additional experiments and ablations.

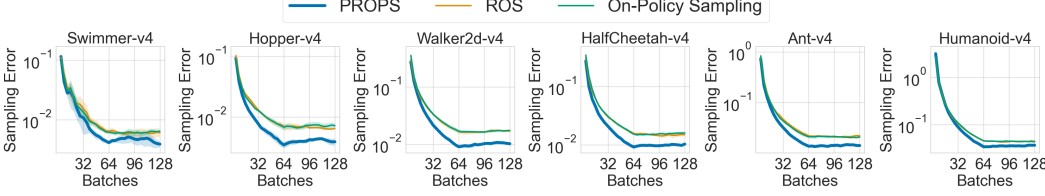

Figure 7: Sampling error with a fixed, expert target policy. Solid curves denote the mean over 5 seeds. Shaded regions denote 95% confidence belts.

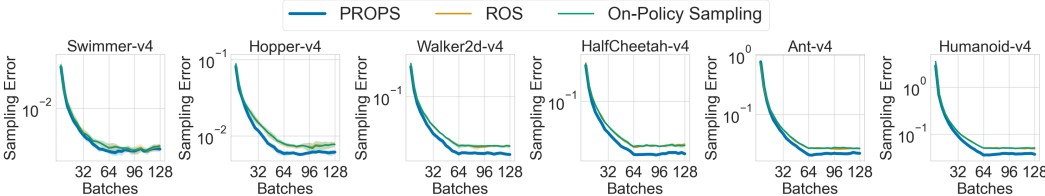

Figure 8: Sampling error with a fixed, randomly initialized target policy. Solid curves denote the mean over 5 seeds. Shaded regions denote 95% confidence belts.

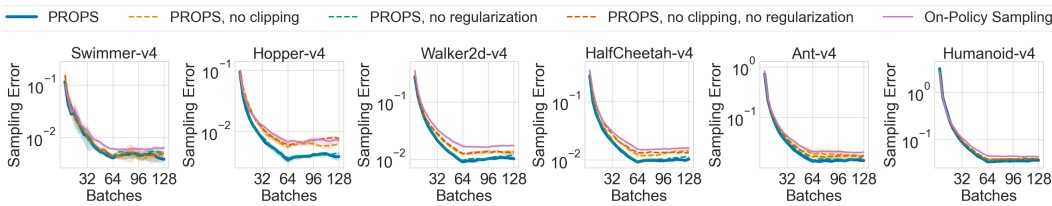

Figure 9: Sampling error ablations with a fixed, expert target policy. Here, "no clipping" refers to setting $\epsilon_{\text{PROPS}} = \infty$, and "no regularization" refers to setting $\lambda = 0$. Solid curves denote the mean over 5 seeds, and shaded regions denote 95% bootstrap confidence intervals.

### D.1 CORRECTING SAMPLING ERROR FOR A FIXED TARGET POLICY

In this appendix, we expand upon results presented in Section 6.1 of the main paper and provide additional experiments investigating the degree to which PROPS reduces sampling error with respect to a fixed target policy. We include empirical results for all six MuJoCo benchmark tasks as well as ablation studies investigating the effects of clipping and regularization.

We tune PROPS and ROS using a hyperparameter sweep. For PROPS, we consider learning rates in $\{10^{-3}, 10^{-4}\}$, regularization coefficients $\lambda \in \{0.01, 0.1, 0.3\}$, and PROPS target KLs in $\delta_{\text{PROPS}} \in \{0.05, 0.1\}$. We fix $\epsilon_{\text{PROPS}} = 0.3$ across all experiments. For ROS, we consider learning rates in $\{10^{-3}, 10^{-4}, 10^{-5}\}$. We report results for the hyperparameters yielding the lowest sampling error.

Fig. 7 and 8 show sampling error computed with a fixed expert and randomly initialized target policy, respectively. We see that PROPS achieves lower sampling error than both ROS and on-policy sampling across all tasks. ROS shows little to no improvement over on-policy sampling, highlighting the difficulty of applying ROS to higher dimensional tasks with continuous actions.

Fig. 9 ablates the effects of PROPS's clipping mechanism and regularization on sampling error reduction. We ablate clipping by setting $\epsilon_{\text{PROPS}} = \infty$, and we ablate regularization by setting $\lambda = 0$. We use a fixed expert target policy and use the same tuning procedure described earlier in this appendix. In all tasks, PROPS achieves higher sampling error without clipping nor regularization than it does with clipping and regularization. However, it nevertheless outperforms on-policy sampling in all tasks except Hopper where it matches the performance of on-policy sampling. Only including regularization slightly decreases sampling error, whereas clipping alone produces sampling error only slightly higher than that achieved by PROPS with both regularization and clipping. These observations indicate that while regularization in is helpful, clipping has a stronger effect on sampling error reduction than regularization when the target policy is fixed.

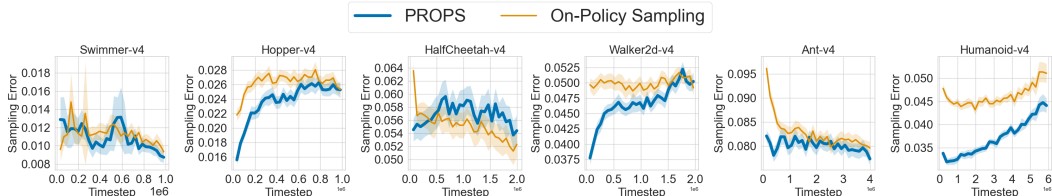

Figure 10: Sampling error throughout RL training. Solid curves denote the mean over 5 seeds. Shaded regions denote 95% confidence belts.

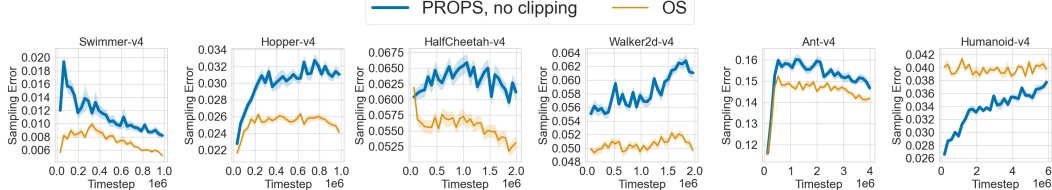

Figure 11: Sampling error throughout RL training without clipping the PROPS objective. Solid curves denote the mean over 5 seeds. Shaded regions denote 95% confidence belts.

### D.2 CORRECTING SAMPLING ERROR DURING RL TRAINING

In this appendix, we include additional experiments investigating the degree to which PROPS reduces sampling error during RL training, expanding upon results presented in Section 6.2 of the main paper. We include sampling error curves for all six MuJoCo benchmark tasks and additionally provide ablation studies investigating the effects of clipping and regularization on sampling error reduction and data efficiency in the RL setting.

As shown in Fig 10, PROPS achieves lower sampling error than on-policy sampling throughout training in 5 out of 6 tasks. We observe that PROPS increases sampling error but nevertheless improves data efficiency in HalfCheetah as shown in Fig. 4a. This result likely arises from our tuning procedure in which we selected hyperparameters yielding the largest return. Although lower sampling error intuitively correlates with increased data efficiency, it is nevertheless possible to achieve high return without reducing sampling error.

In our next set of experiments, we ablate the effects of PROPS's clipping mechanism and regularization on sampling error reduction and data efficiency. We ablate clipping by tuning RL agents with $\epsilon_{\text{PROPS}} = \infty$, and we ablate regularization by tuning RL agents with $\lambda = 0$. Fig. 11 and Fig. 12 show sampling error curves without clipping and without regularization, respectively. Without clipping, PROPS achieves larger sampling than on-policy sampling in all tasks except Humanoid. Without regularization, PROPS achieves larger sampling error in 3 out of 6 tasks. These observations indicate that while clipping and regularization both help reduce sampling during RL training, clipping has a stronger effect on sampling error reduction. As shown in Fig. 13 PROPS data efficiency generally decreases when we remove clipping or regularization.

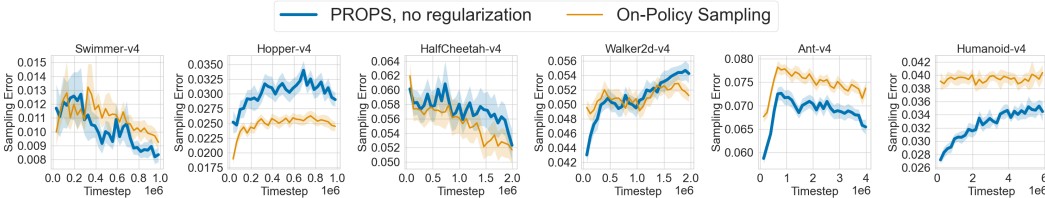

Figure 12: Sampling error throughout RL training without regularizing the PROPS objective. Solid curves denote the mean over 5 seeds. Shaded regions denote 95% confidence belts.

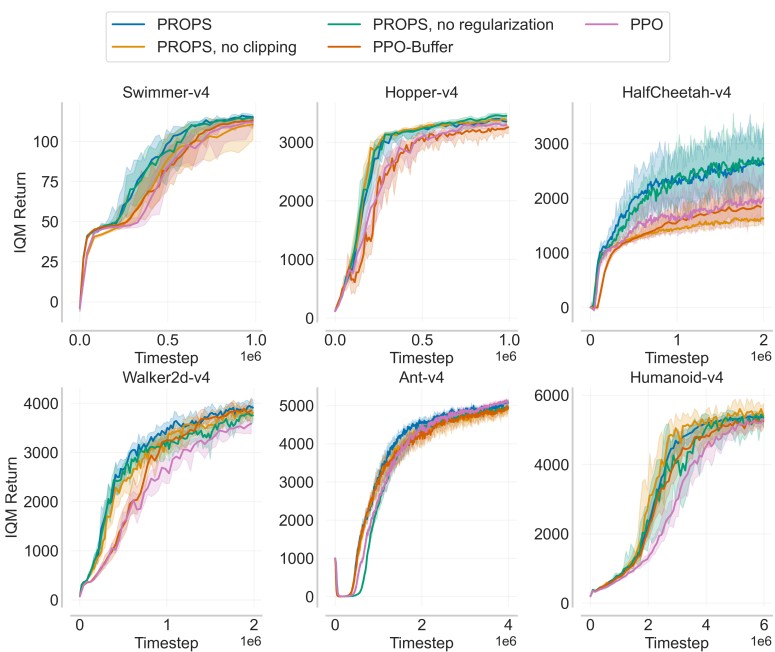

Figure 13: IQM return over 50 seeds of PROPS with and without clipping or regularizing the PROPS objective. Shaded regions denote 95% bootstrap confidence intervals.

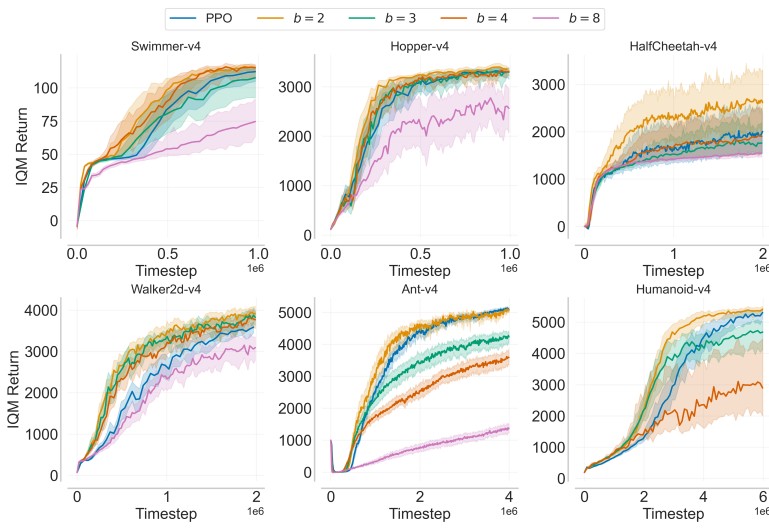

Figure 14: IQM return over 50 seeds for PROPS with different buffer sizes. We exclude $b = 8$ for Humanoid-v4 due to the expense of training and tuning. Shaded regions denote 95% bootstrap confidence intervals.

Lastly, we consider training with larger buffer sizes $b$ in Fig. 14. We find that data efficiency may decrease with a larger buffer size. Intuitively, the more historic data kept around, the more data that must be collected to impact the aggregate data distribution.

## D.3 BIAS AND VARIANCE OF PROPS

In Fig. 15, we investigate the bias and variance of the empirical state-action visitation distribution $d^{\mathcal{D}}(s, a)$ under PROPS, ROS, and on-policy sampling. We report the bias and variance averaged over

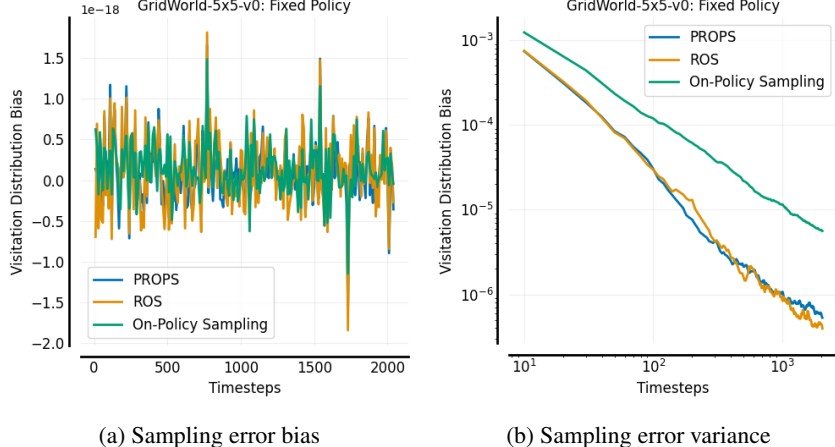

(a) Sampling error bias        (b) Sampling error variance

Figure 15: Sampling error bias and variance estimates of different sampling methods. Empirically, PROPS is unbiased and lower variance than on-policy sampling.

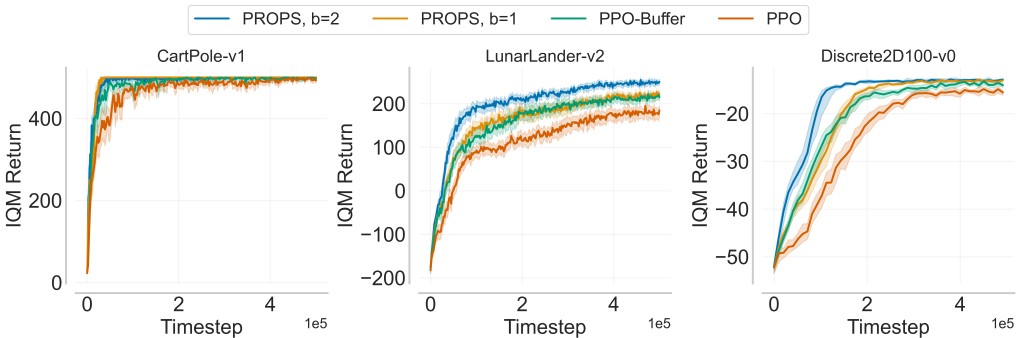

Figure 16: IQM return for discrete action tasks over 50 seeds. Shaded regions denote 95% bootstrap confidence intervals.

all $(s, a) \in \mathcal{S} \times \mathcal{A}$ computed as follows:

$$\text{bias} = \frac{1}{|\mathcal{S} \times \mathcal{A}|} \sum_{(s,a) \in \mathcal{S} \times \mathcal{A}} \left( \mathbb{E}\left[ d^{\mathcal{D}}(s, a) \right] - d_{\pi_\theta}(s, a) \right) \tag{10}$$

$$\text{variance} = \frac{1}{|\mathcal{S} \times \mathcal{A}|} \sum_{(s,a) \in \mathcal{S} \times \mathcal{A}} \mathbb{E}\left[ \left( d^{\mathcal{D}}(s, a) - d_{\pi_\theta}(s, a) \right)^2 \right] \tag{11}$$

As shown in Fig. 15, the visitation distribution under PROPS and ROS empirical have near zero bias (note that the vertical axis has scale $10^{-18}$) and have lower variance than on-policy sampling.

## D.4 DISCRETE-ACTION TASKS

We include 3 additional discrete-action domains of varying complexity. The first two are the widely used OpenAI gym domains CartPole-v1 and LunarLander-v2 (Brockman et al., 2016). The third is a 2D navigation task, Discrete2D100-v0, in which the agent must reach a randomly sampled goal. There are 100 actions, each action corresponding to different directions in which the agent can move. From Fig. 16 and 17 we observe that PROPS with $b = 2$ achieves larger returns than PPO and PPO-BUFFER all throughout training in all three tasks. PROPS with $b = 1$ (no historic data) achieves larger returns than PPO all throughout training in all three tasks and even outperforms PPO-BUFFER in CartPole-v1 and Discrete2D100-v0 even though PPO-BUFFER learns from twice as much data. Thus, PROPS can improve data efficiency *without* historic data.

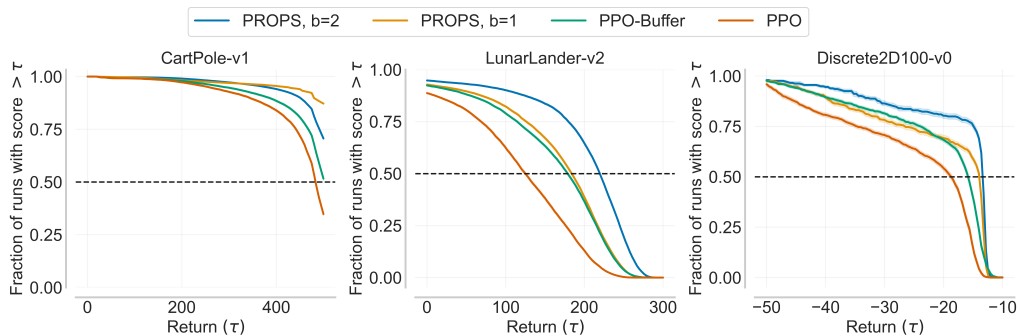

Figure 17: Performance profiles for discrete-action tasks over 50 seeds. Higher values correspond to more reliable convergence to high-return policies. Shaded regions denote 95% bootstrap confidence intervals.

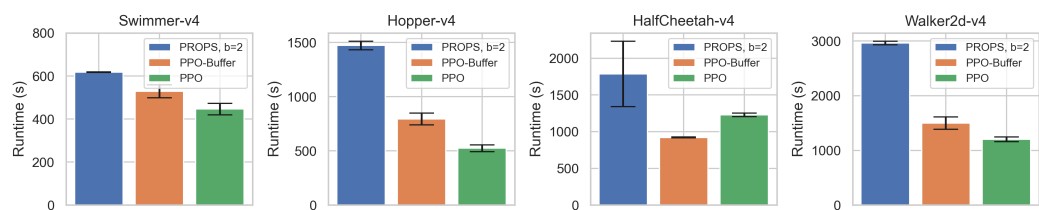

Figure 18: Runtimes for PROPS, PPO-BUFFER, and PPO. We report means and standard errors over 3 independent runs.

### D.5    RUNTIME COMPARISONS

Figure 18 shows runtimes for PROPS, PPO-BUFFER, and PPO averaged over 3 runs. We trained all agents on a MacBook Air with an M1 CPU and use the same tuned hyperparameters used throughout the paper. PROPS takes at most twice as long as PPO-BUFFER; intuitively, both PROPS and PPO-BUFFER learn from the same amount of data but PROPS learns two policies.

We note that PPO-BUFFER is faster than PPO is HalfCheetah-v4 because, with our tuned hyperparameters, PPO-BUFFER performs fewer target policy updates than PPO. In particular, PPO-BUFFER is updating its target policy every 4096 steps, whereas PPO is updating the target policy every 1024 steps.

## E    HYPERPARAMETER TUNING FOR RL TRAINING

For all RL experiments in Section 6.2 and Appendix D.2, we tune PROPS, PPO-BUFFER, and PPO separately using a hyperparameter sweep over parameters listed in Table 2 and fix the hyperparameters in Table 5 across all experiments. Since we consider a wide range of hyperparameter values, we ran 10 independent training runs for each hyperparameter setting. We then performed 50 independent training runs for the hyperparameters settings yielding the largest returns at the end of RL training. We report results for these hyperparameters in the main paper. Fig. 19 shows training curves obtained from a subset of our hyperparameter sweep.

| | |
|---|---|
| PPO learning rate | $10^{-3}, 10^{-4}$, linearly annealed to $0$ over training |
| PPO batch size $n$ | $1024, 2048, 4096, 8192$ |
| PROPS learning rate | $10^{-3}, 10^{-4}$ (and $10^{-5}$ for Swimmer) |
| PROPS behavior batch size $m$ | $256, 512, 1024, 2048, 4096$ satisfying $m \leq n$ |
| PROPS KL cutoff $\delta_{\text{PROPS}}$ | $0.03, 0.05, 0.1$ |
| PROPS regularizer coefficient $\lambda$ | $0.01, 0.1, 0.3$ |

Table 2: Hyperparameters used in our hyperparameter sweep for RL training.

| Environment | Batch Size | Learning Rate |
|---|---|---|
| Swimmer-v4 | 4096 | $10^{-3}$ |
| Hopper-v4 | 2048 | $10^{-3}$ |
| HalfCheetah-v4 | 1024 | $10^{-4}$ |
| Walker2d-v4 | 4096 | $10^{-4}$ |
| Ant-v4 | 1024 | $10^{-3}$ |
| Humanoid-v4 | 8192 | $10^{-4}$ |

Table 3: Tuned PPO hyperparameters

| Environment | PPO Batch Size | PPO Learning Rate | PROPS Batch Size | PROPS Learning Rate | PROPS KL Cutoff | PROPS Regularization $\lambda$ |
|---|---|---|---|---|---|---|
| Swimmer-v4 | 2048 | $10^{-3}$ | 1024 | $10^{-5}$ | 0.03 | 0.1 |
| Hopper-v4 | 2048 | $10^{-3}$ | 256 | $10^{-3}$ | 0.05 | 0.3 |
| HalfCheetah-v4 | 1024 | $10^{-4}$ | 512 | $10^{-3}$ | 0.05 | 0.3 |
| Walker2d-v4 | 2048 | $10^{-3}$ | 256 | $10^{-3}$ | 0.1 | 0.3 |
| Ant-v4 | 2048 | $10^{-4}$ | 256 | $10^{-3}$ | 0.03 | 0.1 |
| Humanoid-v4 | 8192 | $10^{-4}$ | 256 | $10^{-4}$ | 0.1 | 0.1 |

Table 4: Hyperparameters used in our hyperparameter sweep for RL training.

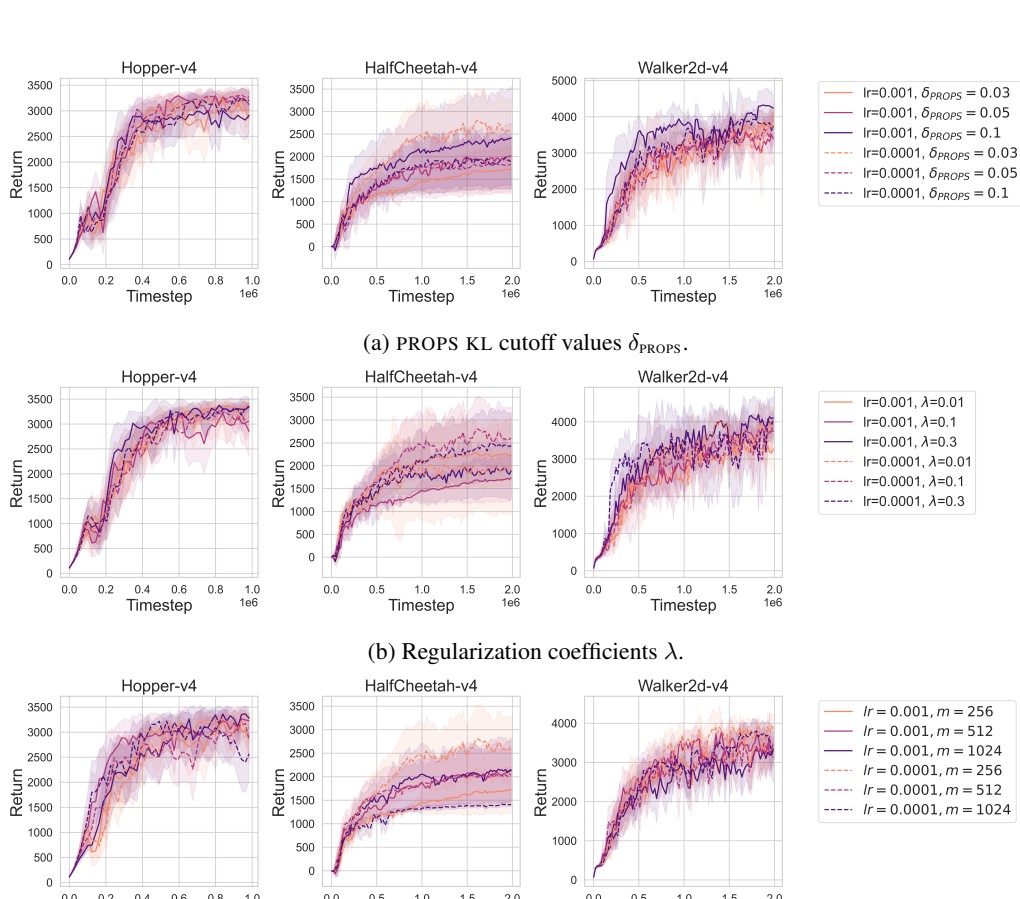

(a) PROPS KL cutoff values $\delta_{\text{PROPS}}$.

(b) Regularization coefficients $\lambda$.

(c) Behavior batch sizes $m$ (*i.e.* the number of steps between behavior policy updates).

Figure 19: A subset of results obtained from our hyperparameter sweep. Default hyperparameter values are as follows: PROPS KL cutoff $\delta_{\text{PROPS}} = 0.03$; regularization coefficient $\lambda = 0.1$; behavior batch size $m = 256$. Darker colors indicate larger hyperparameter values. Solid and dashed lines have the PROPS learning rate set to $1 \cdot 10^{-3}$ and $1 \cdot 10^{-4}$, respectively. Curves denote averages over 10 seeds, and shaded regions denote 95% confidence intervals.

| | |
|---|---|
| PPO number of update epochs | 10 |
| PROPS number of update epochs | 16 |
| Buffer size $b$ | 2 target batches (also 3, 4, and 8 in Fig. 14) |
| PPO minibatch size for PPO update | $bn/16$ |
| PROPS minibatch size for PROPS update | $bn/16$ |
| PPO and PROPS networks | Multi-layer perceptron with hidden layers (64,64) |
| PPO and PROPS optimizers | Adam (Kingma and Ba, 2015) |
| PPO discount factor $\gamma$ | 0.99 |
| PPO generalized advantage estimation (GAE) | 0.95 |
| PPO advantage normalization | Yes |
| PPO loss clip coefficient | 0.2 |
| PPO entropy coefficient | 0.01 |
| PPO value function coefficient | 0.5 |
| PPO and PROPS gradient clipping (max gradient norm) | 0.5 |
| PPO KL cut-off | 0.03 |
| Evaluation frequency | Every 10 target policy updates |
| Number of evaluation episodes | 20 |

Table 5: Hyperparameters fixed across all experiments. We use the PPO implementation provided by CleanRL (Huang et al., 2022).

