# OpenReview forum: "On-Policy Policy Gradient Reinforcement Learning Without On-Policy Sampling"
_ICLR.cc/2025/Conference — Submitted to ICLR 2025_

### Official Review · Reviewer_iwvF · 2024-10-31

**Soundness:** 3
**Presentation:** 3
**Contribution:** 2
**Rating:** 5
**Confidence:** 5

**Summary:**

This paper proposes Proximal Robust On-Policy Sampling (PROPS), an adaptive off-policy sampling method designed to reduce sampling error and improve data efficiency in on-policy policy gradient algorithms. The paper generalizes the adaptive off-policy sampling method ROS introduced in Zhong et al. (2022) by proposing regularization techniques to handle continuous action spaces, and incorporates PROPS into RL training with PPO. Experiments on GridWorld, other discrete-action tasks, and MuJoCo benchmarks compare PPO with PROPS sampling to standard on-policy PPO and a version of PPO that naively reuses recent data without off-policy corrections.

**Strengths:**

- **[S1] Clear and well-written:** The paper is easy to read. The authors clearly present the interesting idea of on-policy data vs. on-policy sampling, as well as the proposed PROPS algorithm.
- **[S2] Detailed, transparent presentation of experimental implementation:** The authors do a nice job of clearly describing the implementation details used in the experiments. This includes supporting analysis in the Appendix showing results across hyperparameter values.

**Weaknesses:**

**[W1] Lack of novel theoretical analysis**
- While Proposition 1 is valuable because it is not obvious that this sampling procedure converges to the correct state visitation distribution, it is a slight generalization of a result that already appears in Zhong et al. (2022). It is also restricted to finite states and actions, and is based on a different sampling procedure than what is proposed in Section 5.
- Proposition 1 only analyzes what happens in the limit of infinite data, but the transient behavior is important in the context of policy gradient algorithms where a finite (and often limited) amount of data is collected between each policy update.
- It seems that there would likely be an important interplay between the size of policy updates, the amount of data to reuse, and the size of behavior policy updates determined by the PROPS objective. Unfortunately, the paper does not perform any analysis to connect these components, and instead sets all hyperparameters using a hyperparameter sweep.

**[W2] Experiments do not demonstrate convincing performance benefits**
- The experimental results suggest that PROPS can achieve slightly better sampling error during RL training, but these improvements only lead to marginal performance benefits in continuous control MuJoCo tasks.
- Experiments only compare against PPO and a version of PPO that naively reuses recent data (PPO-Buffer). There have been several works that improve data efficiency in a more principled way than PPO-Buffer that are similar to the goals of this work (see references in lines 122-124), but none of these are considered as baselines. Off-policy RL algorithms are also not included for comparison, which can significantly outperform on-policy or near on-policy methods in some of the tasks considered in this work.
- Due to the regularization needed for stable performance, PROPS requires the behavior policy to remain close to the current policy (small $\delta_{\textnormal{PROPS}}$). The experimental results suggest that this prevents PROPS from reusing a significant amount of past data (as mentioned in lines 470-471 and 1181-1183), limiting the data efficiency that can be achieved.

**[W3] Contribution is not clear relative to existing methods:** The paper does not clearly position its contribution relative to existing RL methods that also address data or computational efficiency. See [Q1] below.

**Questions:**

**[Q1]** What do the authors believe are the main benefits of PROPS relative to existing methods that address data or computational efficiency, including:
- Methods that reuse data by making off-policy corrections (references in lines 122-124): These algorithms appear to be the most similar to PROPS, and some of them introduce similar levels of added complexity / runtime when compared to on-policy algorithms. However, they are not compared against in the experiments.
- Off-policy RL algorithms: These methods have been shown to be extremely data efficient and can even be used to train complex tasks directly from real-world data (e.g., [1]). They have also been shown to significantly outperform on-policy methods on some of the MuJoCo tasks used in this paper.
- Massively parallel on-policy implementations: When a simulator is available, it has been shown that massively parallel implementations of on-policy algorithms such as PPO can result in very efficient RL training in terms of wall-clock time (e.g., [2]).

**[Q2] Experiments**
- The use of different hyperparameter values across each algorithm makes the comparison of PROPS and PPO difficult to interpret. I appreciate the value in the best vs. best comparisons presented in the paper, but how does PROPS perform when using the same hyperparameter values as PPO (and only tunes the PROPS-specific hyperparameters)?
- It is not obvious to me that reducing sampling error will necessarily lead to improved performance. Given the non-convex objective of policy gradient methods, noisy updates could even be helpful to avoid / escape local optima. Did you perform any experimental analysis on this connection (e.g., comparing PPO for the same number of total updates but varying batch sizes)? A comparison of PPO and PROPS for $b=1$ (with the same PPO hyperparameters) would also provide some insight into this question.
- It seems that “on-policy sampling” refers to PPO-Buffer in the sampling error figures, which does not seem like a fair comparison because PPO-Buffer does not attempt to correct for the use of off-policy data at all. What is the sampling error of standard on-policy PPO (and how does it compare to PROPS for $b=1$ with the same PPO hyperparameters)? Are similar trends observed?

**Minor:**
- In (1), the combination of an expectation over the visitation distribution and a summation over time inside the expectation is a bit strange. When summing over time inside the expectation, the expectation is typically written w.r.t. trajectories rather than the visitation distribution.
- Equation (5) can be written more directly as $\min(-\pi_{\phi} / \pi_{\theta}, -(1-\epsilon_{\textnormal{PROPS}}))$. The more complicated structure currently being used is needed in PPO’s objective because advantages can be positive or negative, but this is not relevant in (5).
- Why is IQM used as the performance metric? Typically the mean return is reported.
- Typos: line 341: Samping -> Sampling, line 485: Figure 5b -> Figure 3b
- Some of the legends appear inconsistent with other figures (Figure 12) / with the figure caption (Figure 15)

---

**References:**

[1] Smith et al. Demonstrating a Walk in the Park: Learning to Walk in 20 Minutes With Model-Free Reinforcement Learning. In RSS 2023.

[2] Rudin et al. Learning to Walk in Minutes Using Massively Parallel Deep Reinforcement Learning. In CoRL 2022.

---

> ### Author Response · Authors · 2024-11-19
>
> Thank you for your thoughtful review! We’re pleased to see that you found our paper clear and well-written and appreciated our full transparency of experiment details and hyperparameters. Below, we answer your questions.
>
> #  [W1] Theoretical Analysis
>
> > [Proposition 1] is a slight generalization of a result that already appears in Zhong et al. (2022). It is also restricted to finite states and actions, and is based on a different sampling procedure than what is proposed in Section 5.
>
> First, thank you for recognizing the value of Proposition 1!
>
> Proposition 1 considers the sampling procedure used by ROS (Zhong et. al 2022), since PROPS reduces to ROS if we only perform a single update on the behavior policy rather than multiple minibatch updates.  To see this, recall that $\phi$ refers to the behavior policy parameters and $\theta$ refers to the target policy parameters, and at the beginning of the PROPS update, we set $\phi \gets \theta$. Then if we perform a single behavior update, we have $\pi_\phi(a|s) / \pi_\theta(a|s) = 1$, and the PROPS update becomes $\nabla E_\pi[-1] = E_\pi[-\nabla \log \pi_\phi(a|s)]$ which is equivalent to the ROS update.
>
> Just as PPO serves as a more scalable approximation of the theoretically motivated TRPO, PROPS serves as a more scalable approximation of the theoretically motivated ROS algorithm.
>
> > Proposition 1 only analyzes what happens in the limit of infinite data, but the transient behavior is important in the context of policy gradient algorithms where a finite (and often limited) amount of data is collected between each policy update
>
> We agree that it would be interesting to quantify the rate at which ROS reduces sampling error as a function of sample size, though statistical consistency is an important property to establish, as we always want to make sure we indeed achieve zero sampling error in the limit of infinite data.
>
> > It seems that there would likely be an important interplay between the size of policy updates, the amount of data to reuse, and the size of behavior policy updates determined by the PROPS objective
>
> We agree that it would be worthwhile to study the interplay between different PROPS hyperparameters, though our focus was ensuring we gave each method the best hyperparameters. We can add additional ablations studying this interplay in a camera ready version.
>
> # [W2] Regarding empirical improvement in data efficiency.
>
> The core goal of our work is to develop nuance in what it means to learn “on-policy” and demonstrate how adaptive off-policy sampling can improve performance in on-policy policy gradient learning. Dramatic improvement on benchmarks is not necessary for providing evidence for the claims we make in this paper.
>
> > The experimental results suggest that [KL regularization] prevents PROPS from reusing a significant amount of past data...limiting the data efficiency that can be achieved.
>
> Enabling PROPS to reuse historic data from many previous updates is an interesting challenge for future work. The core challenge is not necessarily that regularization prevents the behavior policy from changing too much. Retaining data from multiple previous updates increases the total amount of data in the agents buffer $D$, so the agent needs to collect a larger number of samples to adjust the distribution of $D$. The core issue is that the agent is not collecting enough data between updates to make $D$ more closely distributed to on-policy. We plan to address this challenge in future work by letting the agent increase or decrease the number of samples the behavior policy collects between target updates according to how closely $D$ matches the current on-policy distribution.
>
> # [W3/Q1] Main benefits of PROPS relative to existing methods (off-policy corrections, off-policy algorithms, massively parallel training)
>
> We agree that off-policy correction methods and off-policy algorithms are related, though we believe such a comparison would distract from the core ideas of the paper. Our work’s core goal is to develop nuance in the on-policy vs off-policy dichotomy in the context of policy gradient learning, not to propose a new SOTA on-policy policy gradient algorithm.
>
> While some applications permit just increasing the batch size via parallelization, other applications have stricter sample budgets and consequently a large body of RL research attempts to understand how to develop data efficient learners (e.g. [1-3]). PROPS is more closely related in spirit to the latter line of work. We can emphasize this point in our revisions.
>
> [1] Janner et. al. When to Trust Your Model: Model-Based Policy Optimization. NeurIPS 2019. https://arxiv.org/abs/1906.08253
>
> [2] Chen et. al. Randomized Ensembled Double Q-Learning: Learning Fast Without a Model. ICLR 2021. https://arxiv.org/abs/2101.05982
>
> [3] D'Oro et. al. Sample-Efficient Reinforcement Learning by Breaking the Replay Ratio Barrier. ICLR 2023. https://openreview.net/forum?id=OpC-9aBBVJe

---

> ### Author Response · Authors · 2024-11-20
>
> # [Q2] Experiments
>
> ## PROPS hyperparameters
>
> > how does PROPS perform when using the same hyperparameter values as PPO (and only tunes the PROPS-specific hyperparameters)
>
> In our current Hopper-v4 and Humanoid-v4 experiments, PROPS uses the same batch size and learning rate as PPO, and we see improvements in data efficiency. (Hyperparameters for each experiment are included in the `commands` directory provided in our supplemental material – we will add these to the appendix).
>
> ## Does increasing the batch size improve performance? (Do we actually want to reduce sampling error?)
>
> > It is not obvious to me that reducing sampling error will necessarily lead to improved performance. Given the non-convex objective of policy gradient methods, noisy updates could even be helpful to avoid / escape local optima.
>
> It is possible that true policy gradient may not help with non-convexity, however, on-policy policy gradient algorithms operate under the assumption that it is desirable to follow the true policy gradient. Policy updates are more accurate when our data more closely matches the on-policy distribution, so under this assumption, reducing sampling error should in principle improve performance.
>
> > Did you perform any experimental analysis on this connection (e.g., comparing PPO for the same number of total updates but varying batch sizes)?
>
> We attach two figures for a few representative tasks from our PPO parameter sweep:
>
> [Fig. 1 (link)](https://drive.google.com/file/d/1Z9nmiSfXmR0zFqiPhc6d89P9saH6SoYn/view?usp=share_link): PPO returns vs. the number of updates performed for various batch sizes (10 seeds). Note that in our experiments, we fixed the training budget for each task, so runs with larger batch sizes perform fewer updates. This figure shows that increasing the batch size yields greater returns in fewer updates. Since larger batches have lower sampling error, reduced sampling error does indeed correspond to better performance on MuJoCo tasks.
>
> [Fig. 2 (link)](https://drive.google.com/file/d/1U0Mejf5acw__BoAVz_rfSKFGM2OeT5xZ/view?usp=share_link): PPO returns vs. the number of timesteps (10 seeds). This figure shows that  increasing the batch size *sometimes* yields greater returns in fewer *timesteps* (e.g. PPO with a batch size of 2048 yields the most data efficient learning on Hopper-v4). In our work, we care about *data efficiency*, the number of *samples* required to achieve high returns within a fixed training budget, so we select the batch size whose training curve lies above all others on the return vs. timesteps figure. The most data efficient batch size is not necessarily the most update-efficient batch size.
>
> ## Sampling error experiments: comparing PROPS with PPO-Buffer vs PPO.
>
> > What is the sampling error of standard on-policy PPO (and how does it compare to PROPS for b=1 with the same PPO hyperparameters)? Are similar trends observed?
>
> We made this comparison in our sampling error experiments with a fixed target policy in Section 6.1. More concretely, comparing PROPS with b=1 (when there is no historic data in the agent’s buffer) to vanilla PPO would assess if PROPS reduces sampling error w.r.t a fixed target policy, and that is precisely what we assess in section 6.1.
>
> > It seems that “on-policy sampling” refers to PPO-Buffer in the sampling error figures, which does not seem like a fair comparison because PPO-Buffer does not attempt to correct for the use of off-policy data at all.
>
> When computing sampling error in our RL experiments, we compare PROPS with b=2 to PPO-Buffer with b=2 rather than vanilla PPO to assess if PROPS reduces sampling error in historic data (i.e. if PROPS reduces sampling error w.r.t a changing target policy).
>
> Please let us know if our clarification on this point answers your question. We can provide further explanation if needed!
>
> # Minor comments
>
> > In (1), the combination of an expectation over the visitation distribution and a summation over time inside the expectation is a bit strange.
>
> Thank you for pointing this out! We will rewrite it as an expectation over trajectories in our revisions.
>
> > Equation (5) can be written more directly as
> min(−πϕ/πθ,−(1−ϵPROPS))
>
> Since the PROPS objective (Eq. 5) is equivalent to the PPO objective (Eq. 3) with the advantage A(s,a) is set to -1 for all (s,a), we write the PROPS objective using the same form as the PPO objective. We agree it can be written more simply (e.g. the OpenAI Spinning Up documentation for PPO links to a simplified formulation similar to what you suggest [here](https://drive.google.com/file/d/1PDzn9RPvaXjJFZkGeapMHbHGiWWW20Ey/view?pli=1)).

---

> > ### Author Response · Authors · 2024-11-20
> >
> > > Why is IQM used as the performance metric? Typically the mean return is reported.
> >
> > Agarwal et. al [1] propose the IQM as a more robust alternative to the mean, and this metric has been used in many recent RL works (e.g, [2-4]).
> >
> > [1] Agarwal et. al. Deep Reinforcement Learning at the Edge of the Statistical Precipice. NeurIPS 2021. https://arxiv.org/abs/2108.13264
> >
> > [2] Sokar et. al. The Dormany Neuron Effect in Deep Reinforcement Learning. ICML 2023. https://proceedings.mlr.press/v202/sokar23a/sokar23a.pdf
> >
> > [3] Nikishin et. al. Deep Reinforcement Learning with Plasticity Injection. NeurIPS 2023. https://arxiv.org/pdf/2305.15555
> >
> > [4] Schwarzer et. al. Pretraining Representations for Data-Efficient Reinforcement Learning. NeurIPS 2021 https://arxiv.org/pdf/2106.04799
> >
> > > [A few typos]
> >
> > Thank you for pointing out these typos! We will fix these in our revisions.
> >
> > ---
> >
> > Thank you again for leaving a detailed review! Please let us know if we have clarified your questions. If you have follow-up questions, we'd be happy to discuss further!

---

> > > ### Comment · Reviewer_iwvF · 2024-11-22
> > > **Response to Authors**
> > >
> > > Thank you for your detailed responses. Unfortunately, I do not believe that the contributions of the paper are significant enough to be suitable for publication in its current form.
> > >
> > > I do not believe that “developing nuance in the on-policy vs off-policy dichotomy” represents a significant enough contribution on its own, unless it is supported by strong theoretical analysis or convincing empirical performance. This is especially true given that the prior work from Zhong et al. (2022) has already established this nuance in the policy evaluation setting, and this work applies that nuance in the context of policy gradient algorithms.
> > >
> > > **Theory:**
> > >
> > > - Theoretical results should be established that support the claim of data efficiency. Establishing consistency is important but is not sufficient theoretical analysis, especially in the setting considered in this paper where PROPS is being used to correct relatively small batch sizes at every policy update.
> > > - I mentioned in my review that there is an important interplay between the size of policy updates, the amount of data to reuse, and the size of behavior policy updates determined by the PROPS objective. To clarify, I believe this should be analyzed from a theoretical perspective and then supported by empirical results, not from an empirical ablation study. If you can automatically set these parameters based on theoretical results that provide provable benefits compared to on-policy algorithms in some way while retaining the same approximate policy improvement guarantees, that would be an interesting contribution.
> > >
> > > **Experiments:**
> > >
> > > - Based on my understanding of the results in Section 6.1 (please let me know if I have misunderstood), they are not the same as considering the $b=1$ case during RL training. Section 6.1 evaluates a much larger replay buffer size for a given target policy (combined over 64 batches in Figures 8-9). For $b=1$ during RL training, the buffer size would be considerably smaller. PPO with $b=1$ should be added for comparison in the sampling error analysis, since PPO-Buffer introduces a distribution shift without any correction.
> > > - PROPS should be compared against off-policy correction methods in experiments, which I believe are the most closely related to the goals of PROPS. These algorithms typically do not have state-of-the-art performance, but they have theoretical support and have shown data efficiency improvements w.r.t. on-policy algorithms. In order to demonstrate why PROPS is useful, I believe it should outperform this class of algorithms.
> > > - The authors discuss the goal of data efficiency in their responses, but I suspect it will be very difficult for PROPS to match the data efficiency of off-policy algorithms with high update-to-data (UTD) ratios that have shown very strong performance in recent years (the authors include references to examples of these algorithms in their responses). For this reason, there is probably more opportunity to develop the theoretical foundation of PROPS, and demonstrate that PROPS outperforms other methods with similar theoretical support like off-policy correction techniques.

---

> ### Author Response · Authors · 2024-12-04
>
> We appreciate the continued discussion! Below, we address your followup comments and questions.
>
> # Contributions over Zhong et. al (2022)
>
> Zhong et. al first established this nuance with ROS, though their empirical analysis is very limited: (1) they only focused on policy evaluation in low dimension tasks like Gridworld and Cartpole, and (2) they showed little to no improvement over on-policy sampling in even low-dimensional continuous policy evaluation tasks. Our work shows how to use the concepts introduced by Zhong et. al to develop a new sampling error correction algorithm that can scale to higher dimensional continuous control settings. We ask the reviewer to reconsider this evaluation on the significant improvement compared to Zhong et al.
>
> # Theory on how different PROPS hyperparameters related to each other
>
> We provide theoretical intuition on the relationship between the amount of historic data reused, the size of target policy updates, and the size of PROPS behavior policy updates in Appendix A of our revised submission. We discuss the following relationships:
>
> 1. When retaining more and more (historic) data, the behavior policy must collect more and more data to make the agent’s data buffer match the on-policy distribution.
> 2. When sampling error is large, behavior policy updates must also be large.
>
> # Sampling error experiments with smaller batch sizes
>
> [Figure 1: Sampling error with a randomly initialized target policy](https://drive.google.com/file/d/1WQevMm_lalUgq-mx-UjcewflyaZxuVp1/view?usp=share_link)
>
> [Figure 2: Sampling error with an expert-quality target policy](https://drive.google.com/file/d/1UsgRRah7moJ5FZ5BvjSY_QCDUNBlF6_s/view?usp=share_link)
>
> Thank you for the clarification! We better understand your comment now. PROPS can reduce sampling error with a small buffer size too. In figures linked above, we zoom in on Figures 7 and 8 (sampling error with a fixed target policy) to more clearly show sampling error after collecting a relatively small number of samples. For clarity, we changed the x axis to the number of transitions in the agent’s buffer.
>
> # Additional baselines
>
> > PROPS should be compared against off-policy correction methods in experiments
>
> Our goal with this work is to improve on-policy policy gradient learning and emphasize a nuance in what it means to learn on-policy, and we designed our experiments to elucidate this nuance. While we agree it would be interesting to compare to off-policy correction methods, our current experiments support the claims we make in our paper. PROPS can in principle be used in addition to off-policy correction methods like Regression Importance Sampling [1], so it could also be interesting to (1) use PROPS to reduce sampling error during data collection, and then (2) use an appropriate off-policy correction method to adjust for any remaining sampling error. However, since our current paper is studying the data collection problem, we leave this extension to future work.
>
> [1] Hanna et. al. Importance Sampling Policy Evaluation with an Estimated Behavior Policy. ICML 2019.
>
> > I suspect it will be very difficult for PROPS to match the data efficiency of off-policy algorithms with high update-to-data (UTD) ratios that have shown very strong performance in recent years
>
> We agree that it will be a challenge for PROPS to match the data efficiency of off-policy algorithms – especially those that can use high UTD ratios. However, our goal with this work is to improve on-policy policy gradient learning and emphasize a nuance in what it means to learn on-policy, not to develop an algorithm that outperforms off-policy algorithms. We agree it would be interesting to understand the extent to which adaptive sampling can close the gap between the data efficiency of on- and off-policy algorithms, though this understanding is not critical to supporting the claims we make in this work.

---

### Official Review · Reviewer_gv5j · 2024-11-03

**Soundness:** 3
**Presentation:** 3
**Contribution:** 3
**Rating:** 8
**Confidence:** 4

**Summary:**

This paper tackles the usual problem of the high-sample complexity of on-policy policy gradient algorithms by refining a very interesting idea proposed by Zhong et al. in 2022. In that paper, the authors observe that when the number of samples is scarce, the samples might resemble the on-policy distribution poorly, which would be equivalent, in a sense, to dealing with an off-policy distribution. In more simple terms, a low number of samples usually cause a high variance policy gradient estimation.

Zhong et al. observe that on-policy samples do not necessarily need to come from the on-policy distribution; for example, the samples

x_i = 0, 1, 1, 0

can come from p(x=1) = 0.5 and p(x=0)=0.5, but also from q(x=1) = 0.4 and q(x=0) = 0.6.
Similarly, a batch of samples that look very off-policy might actually have been generated from the on-policy distribution, for example

x_i = 1, 1, 1, 1,

might have been generated from the policy $p(x=0) = 0.5$, $p(x=1) = 0.5$.
Monte-Carlo estimation (i.e., $E[X] \approx 1/n \sum_i X_i$) has higher variance exactly due to the issue described above, when the number of samples $n$ is low, while the variance disappears for $n\to\infty$, making it a consistent estimator.

The variance is the main problem affecting MC estimators and, thus, policy gradient algorithms as well (which are all MC estimators, as they attempt to estimate a respected value using $1/n \sum_i X_i$).

The variance of MC estimators can be ideally reduced if small batches of samples seem to better adhere to the target distribution. This can be achieved by breaking the i.i.d sampling assumption. For example, when the target distribution is $p(x=0) = 0.5$, $p(x=1) = 0.5$, One can choose a deterministic policy that alternates 0s and 1s. Such policy makes the MC estimator strictly more efficient. In this sense, **an off-policy distribution (that violates the i.i.d assumption) can generate better samples "on-policy" samples than the target distribution.**

Zhong et al. in 2022 propose to learn a policy that aims to lower the variance of the policy gradient estimators by generating samples that adhere better to the target policy distribution.

The contribution of this ICLR 2025 paper is to use the idea of Zhong et al. for on-policy policy gradient estimation (and policy improvement), while in the original publication, the idea was only proposed for policy evaluation.

**Strengths:**

Originality
-------------

The idea proposed is novel. Although the method proposed until page 5 is from Zong's paper, and the application to policy gradient estimation seems somewhat trivial, I think it is still valuable and necessary.

Quality
----------

The quality of the paper is really good. The authors explain the problem and the main idea very well. The method is sounds and directly addresses the problem presented. The experiments are well-designed, explained, and commented on. The results are equipped with good statistical significance, and the appendix reports the hyperparameters necessary for reproducibility.

Clarity
--------

The paper is well-written, and the overall idea is exposed very clearly.

Significance
-----------------

It is hard to assess the significance of this paper: 1) it heavily relies on a previous publication, and 2) it does not seem that the method improves so much over the baselines. However, I think the core idea is significant as it can serve as a source of inspiration for other RL problems, as MC is a core technique in RL to estimate many different quantities.

**Weaknesses:**

Contribution
-----------------

The paper's contribution starts on page 5: the author focuses on presenting Zhong et al.'s paper there. The core contribution is the application of Zhon's idea for PG estimation and the use of PPO clipping and KL divergence to prevent too aggressive updates.

While I think it is necessary that the authors devote that space to expose Zhong et al.'s idea, I think that it might not be clear to the reader that **that is not** the main core idea of the paper.

Clarity
--------

While the paper is well written overall, I do not always like the choice of words and how the problem is described.
I find talking about on-policy "data" vs. "sampling" confusing. A sample is not per se on or off-policy. When we speak about "on-policy" and "off-policy" in estimation, we refer to the generating process (i.e., the distribution) that generates the samples. A sample X_i can be generated by an off-policy distribution $\beta$ and simultaneously very unlikely w.r.t. $\beta$ and very likely w.r.t. the target distribution $\pi$, but that does not make the sample on-policy (or off-policy). Policy gradient estimation is simply off-policy when the distribution that is used to generate the samples is off-policy.

Pairwise, I find speaking of "empirical policy" as in Proposition 1 confusing. One thing is samples, and one thing is the distribution. What can be told, in general, is that if $n > m$, then $Var[1/n \sum_i^n X_i] < Var[1/m \sum_i^m X_i]$, where X_i are samples coming from the same distribution. Now, it would be possible to introduce a new distribution (as done in Proposition 1), and show that the variance of the batch of samples decreases even faster in $n$, when compared to the original distribution.

Overall, I am conscious that much of the "nomenclature" is borrowed from Zhong et al., thus the authors might have preferred to keep consistency with that.

Minor Comments
-----------------------

Equation 1 is unclear as it is written: s_0 comes from a different distribution than s_1, which comes from a different distribution than s_2, and so on. Writing $s \sim d_\pi$ gives the impression that all states are drawn from the same distribution. Usually, there are two ways to express the policy's return,

1.

$$
J(\theta) = \mathbb{E}_{s \sim d^\gamma_\pi, a \sim \pi}\left[r(s, a)\right]
$$

 where $d^\gamma_\pi$ is the discounted state-action visitation (see Nota and Thomas "Is policy gradient a gradient?" and Sutton et al. "Policy gradient with function approximation")

2.

$$
J(\theta) = \mathbb{E}\left[\sum_{t=0}^\infty \gamma^t r(s_t, a_t)\right]
$$

where $s_0 \sim d_0, a_t \sim \pi(\cdot | s_t), s_{t+1} \sim p(\cdot | s_t, a_t)$.

Equation 2 and 4, should be with the discounted state visitation (see Nota and Thomas "Is policy gradient a gradient?" and Sutton et al. "Policy gradient with function approximation" :D )

References
---------------

Nota, C., & Thomas, P. S. (2019). Is the policy gradient a gradient?. arXiv preprint arXiv:1906.07073.

Sutton, R. S., McAllester, D., Singh, S., & Mansour, Y. (1999). Policy gradient methods for reinforcement learning with function approximation. Advances in neural information processing systems, 12. (**see page 3, definition of $d^\pi$**)

**Questions:**

I have only one question:

In Zhong et al., the policy is updated at every step in an online fashion: to me, this makes a lot of sense, since we want constantly to "correct" the sampling distribution. Ideally, one wants to employ the method you describe in Proposition 1.

However, if I understand well, you propose first to use the target policy to collect the samples and, afterward, to generate a policy that "corrects" the previous behavior. However, such a policy is not well defined: it depends on how many samples will be drawn from such a policy. For example, suppose that the target policy is the distribution $\pi(a=0|s) = 0.5$ and $p(a=1|s)=0.5$. Suppose then that the samples drawn from this distribution are:

0, 0, 0, 0, 0.

The policy that you want to learn should then put much more probability density on 1 to compensate, but if we sample hundreds of samples with this new policy, then the dataset will be again "unbalanced". In my understanding, the idea of Zhong et al. to continuously update the policy is to avoid this issue.

Why did you propose this "batch update" rather than an "online update"?

---

> ### Author Response · Authors · 2024-11-18
>
> Thank you for the positive review! We are quite pleased to see that you found our paper  high-quality, well-written, and valuable to the RL community. Below, we answer your questions about PROPS and our notation.
>
> # Clarifying what we mean by on-/off-policy data + Notation changes
>
> > A sample is not per se on or off-policy. When we speak about "on-policy" and "off-policy" in estimation, we refer to the generating process (i.e., the distribution) that generates the samples.
>
> We broadly agree with your comment that a sample is not on- or off-policy per se. Nevertheless, we find it common in the RL literature to confuse the generating process and the samples and chose terminology that we believed would call this out and advance a more nuanced understanding. We agree that it might have been better to say “data sampled off-policy.” For brevity, it may make more sense to keep our current phrasing and add text in the preliminaries section where we note that “off-policy data” is shorthand for “data sampled by off-policy distribution.”
>
> > Overall, I am conscious that much of the "nomenclature" is borrowed from Zhong et al., thus the authors might have preferred to keep consistency with that.
>
> Much of our notation is indeed borrowed from Zhong et. al, though we think it’s reasonable to adjust our notation based on the points you’ve made in your review.
>
> > Equation 2 and 4, should be with the discounted state visitation
>
> Thank you for sharing this reference! In our revisions, we will replace $d_\pi$ with $d^\gamma_\pi$ and note the points raised by Nota and Thomas.
>
>
> # Why we use batched behavior policy updates (rather than updating the behavior policy at every step)
>
> > In Zhong et al., the policy is updated at every step in an online fashion: to me, this makes a lot of sense, since we want constantly to "correct" the sampling distribution...However, if I understand well, you propose first to use the target policy to collect the samples and, afterward, to generate a policy that "corrects" the previous behavior. However, such a policy is not well defined: it depends on how many samples will be drawn from such a policy.
>
> > Why did you propose this "batch update" rather than an "online update"?
>
> This is a good question, and we are planning to address this point in future work.
> You are correct that the optimal behavior policy for a given dataset $D$ depends on the number of samples in $D$ as well as how many samples we will collect with our behavior policy. For simplicity, consider a tabular setting. Let $d_\pi(s,a)$ denote the expected fraction of times $\pi$ observes $(s,a)$, let $d_D(s,a)$ denote the fraction of times $(s,a)$ appears in dataset $D$, and let $d_\beta(s,a)$ denote the expected fraction of times our behavior policy observes $(s,a)$. If we collect $n$ additional samples with our behavior policy and add them to $D$, we want $d_b(s,a)$ to satisfy
> $$
> d(s,a) = \frac{|D| d_D (s,a) + n d_\beta(s,a)}{|D|+n}
> $$
> Updating the behavior policy at every timestep would be computationally expensive, so we propose batched updates to make PROPS more scalable. In principle, PROPS will reduce sample error faster if we reduce the number of steps between behavior policy updates, but doing so would come with a computational cost.
>
> ---
> Thank you again for your positive review, and please let us know if we have addressed your comments and questions! We’ll be happy to discuss further.

---

> > ### Comment · Reviewer_gv5j · 2024-11-26
> > **Thanks**
> >
> > Dear authors, thanks for your rebuttal! I've carefully read your comments. I have no further questions to ask.

---

> > > ### Author Response · Authors · 2024-12-04
> > >
> > > Thank you again for your comments and questions! We appreciate your support!

---

### Official Review · Reviewer_uFws · 2024-11-03

**Soundness:** 2
**Presentation:** 3
**Contribution:** 3
**Rating:** 6
**Confidence:** 4

**Summary:**

The authors tackle the problem of sampling errors from insufficient data such that the empirical data has some discrepancy with desired policy. The authors propose a method to adaptively adjust the policy to resolve this discrepancy. They show their method works in several test cases including control and continuous action space.

** Post rebuttal change **
Now that I understood the paper better and following the discussions I have decided to increase my score.

**Strengths:**

1. The sampling errors problem the authors tackle is interesting, there has not been sufficient discussion on this topic in the RL community.

2. The presentation and writing is generally clear.

3. The sampling errors problem can arise in many real-world setups.

4. The proposes solution is simple in a good way.

**Weaknesses:**

1. The experiments and motivation the authors provided are not accommodating for the problem. When is this really a problem in the real-world? In most standard examples there is ample time to correct the sampling errors with more data and simulation (this happens with a not a lot of samples in practice). Perhaps in cases of non-stationarity of the system itself this "quicker" adaptation become crucial? or maybe for very large state\action-spaces this is more of a problem.

I propose the authors will try to distill where they're problem is really important and design experiments correspondingly, or at least motivate more strongly towards these cases.

2. Even in a quick look, Equation (5) can be rewritten much simpler to be -max(g, 1-eps). I'm not sure forcing it to look like PPO helps with insight, but it is confusing and unnecessary. That is, unless you formulate it similarly to the advantage.

3. Some questions were raised for me regarding the soundness of the proposed solution, see in the questions section.

**Questions:**

1. As a follow-up insight, there may be cases where we can improve the sampling error in one state, but this will cause worse sampling errors down the road because we will be forced to go into specific states. Wouldn't it make more sense to push this sampling error into the reward instead, and form a suitable policy? if you think this is not an issue, it would be helpful if you can explain why.

2. Something doesn't sit right with me regarding equation 6 and its motivation. If we are only aiming to improve the sampling, in the discrete case we wouldn't care about the distance from the target policy. If we are aiming to also improve control, then most algorithms prevent policy changing too much to improve stability in which case we don't need this additional term (for example in PPO). So is this specifically for the continuous case? It does not make a lot of sense to me, at least according to the examples for the discrete case. I would be happy if you could elaborate on this point.

3. Why did you chose the form given in Equation 5? why not its log or difference? Some explanation, ideally guided by theory would be helpful (it is the importance sampling term, widely used in off-policy learning, so it should make some sense using it in some variation).

---

> ### Author Response · Authors · 2024-11-18
>
> We’d like to thank the reviewer for their thoughtful review! We’re glad you found our writing clear and appreciate you recognizing how our work tackles a problem of interest to the RL community! We believe your comments and questions regarding our PROPS algorithm can be addressed with a few minor revisions to our submission.
>
> # When is PROPS useful?
> > When is this really a problem in the real-world? In most standard examples there is ample time to correct the sampling errors with more data and simulation
>
> While some applications permit just increasing the batch size, other applications have stricter sample budgets and consequently a large body of RL research attempts to understand how to develop data efficient learners (e.g. [1-3]). Having said that, we’d like to emphasize that the core contribution of our work is developing nuance in the common usage of on- vs off-policy in the context of policy gradient learning.
>
> [1] Janner et. al. When to Trust Your Model: Model-Based Policy Optimization. NeurIPS 2019. https://arxiv.org/abs/1906.08253
>
> [2] Chen et. al. Randomized Ensembled Double Q-Learning: Learning Fast Without a Model. ICLR 2021. https://arxiv.org/abs/2101.05982
>
> [3] D'Oro et. al. Sample-Efficient Reinforcement Learning by Breaking the Replay Ratio Barrier. ICLR 2023. https://openreview.net/forum?id=OpC-9aBBVJe
>
> # Reducing sampling error at future states
>
> >  there may be cases where we can improve the sampling error in one state, but this will cause worse sampling errors down the road because we will be forced to go into specific states.
>
> This is a great point to discuss; it is possible that reducing sampling error at one state may increase sampling error at future states, and we plan to address this challenge in future work. To ensure we reduce sampling error at the current state and future states, the behavior policy update would need to consider the task’s transition dynamics. While we think it is possible that this challenge could arise, we show empirically in the gridworld domain that PROPS still produces the correct empirical distribution of state-action pairs more efficiently than on-policy sampling.
>
> > Wouldn't it make more sense to push this sampling error into the reward instead, and form a suitable policy?
>
> Could you please clarify what is meant by “push this sampling error into reward”?
>
> # KL regularization for continuous-action tasks.
>
> >  is [the KL regularization in Eq. 6] specifically for the continuous case?
>
> Yes, the KL regularizer in Eq. 6 aims to prevent destructively large behavior policy updates in tasks with continuous actions. PROPS will increase the probability of under-sampled actions, but because the tails of a Gaussian policy are generally under-sampled, the PROPS update may push the behavior policy too far away from the target policy in an effort to over-sample the tails.
>
> # Why we write the PROPS objective in the same form as the PPO objective.
>
> >  Equation (5) can be rewritten much simpler to be -max(g, 1-eps). I'm not sure forcing it to look like PPO helps with insight
>
> > Why did you chose the form given in Equation 5? why not its log or difference?
>
>
> Since the PROPS objective (Eq. 5) is equivalent to the PPO objective (Eq. 3) with the advantage A(s,a) is set to -1 for all (s,a), we write the PROPS objective using the same form as the PPO objective. We agree it can be written more simply (e.g. the OpenAI Spinning Up documentation for PPO links to a simplified formulation similar to what you suggest, see [here](https://drive.google.com/file/d/1PDzn9RPvaXjJFZkGeapMHbHGiWWW20Ey/view?pli=1)). Below, we provide further explanation for the PROPS objective.
>
> To see why PPO (and analogously, PROPS) use the policy ratio rather than its log or difference, recall that on the first policy update where $\theta = \theta_{old}$, the clipped objective reduces to the advantage A(s,a), and the PPO update thus reduces to the vanilla policy gradient update $\nabla E_\pi[A(s,a)] = E_\pi[A(s,a) \nabla \log \pi_\theta(a|s)]$. Thus, PROPS can be viewed as a policy gradient algorithm that decreases the probability of observed action in proportion to how many times they were observed (i.e. actions that were observed more frequently are pushed down more heavily).
>
> Furthermore, If we only perform a single update on the behavior policy rather than multiple minibatch updates, PROPS reduces to the ROS update it builds upon. In this case, $\phi = \theta$ so that $\pi_\phi(a|s) / \pi_\theta(a|s) = 1$. The PROPS update then becomes $\nabla E_\pi[-1] = E_\pi[-\nabla \log \pi_\theta(a|s)]$ which is equivalent to the ROS update. (Recall that $\phi$ refers to the behavior policy parameters and $\theta$ refers to the target policy parameters, and at the beginning of the PROPS update, we set $\phi \gets \theta$).
>
> ---
> Please let us know if this explanation clarifies our formulation of the PROPS update and its connection to the PPO update! If you have further questions, we’d be happy to discuss further.

---

> > ### Comment · Reviewer_uFws · 2024-11-19
> > **Thank you for your reply**
> >
> > I think I understood things a lot better now (though would have been better to understand them while reading the paper initially).

---

> > > ### Author Response · Authors · 2024-12-04
> > >
> > > We're glad we could clarify your comments and address your questions!  In our revisions, we’ve clarified the importance of PROPS’s KL regularization in continuous action tasks (footnote 3 on page 6) and more clearly show how the PROPS objective relates to the PPO objective (lines 297 - 307).
> > >
> > > We ask the reviewer to reconsider their evaluation based on these clarifications during the reviewer/AC Discussion period.

---

### Official Review · Reviewer_8GpQ · 2024-11-04

**Soundness:** 2
**Presentation:** 2
**Contribution:** 2
**Rating:** 3
**Confidence:** 5

**Summary:**

This paper studies the problem that on-policy sampling may have sample error throughout the sampling process. Such sampling errors could be remedied by sampling using a different policy. This paper applied this idea to PPO by remediating the sampling error inside each minibatch. Specifically, this paper divides the minibatch of the PPO into more fine-grained nano batches and adjusts the sampling policy based on sampled nano batches to encourage under-sampled samples. This paper claims their results are better than PPO.

**Strengths:**

Strengths:
1. Estimating the on-policy gradient requires on-policy samples. Because we do not have access to the true gradient, using the target policy to draw on-policy samples to estimate the true gradient has an estimation error. This error is caused by under-sampled or over-sampled data. Encouraging under-sampled on-policy samples reduces the sampling error and improves gradient estimation. This paper is the first to apply this idea to the PPO algorithm.
2. This paper proposes an algorithm to learn the behavior policy to encourage data in PPO algorithms.

**Weaknesses:**

Weaknesses:
1. This paper adjusts the batch size of PPO to "1024,2048,4096,8192" which is much larger than the original batch size (64 samples) of PPO. This paper makes this change because they need to learn a behavior policy to encourage under-sampled data at each update step. In other words, they run a mini-PPO to encourage under-sampled data by finding a behavior policy. This drastic increase in batch size is not well-justified. In fact, it is confusing that their PPO algorithm still achieves the comparable performance of the original PPO with at least 16 times less update due to the batch size increase. I am not sure if the original PPO paper has such large room to change the batch size to 16 times larger while achieving the same performance under the same hyperparameters.
2. This paper is not well-written. The length of this paper can be greatly reduced. Currently, this paper has too many unnecessary examples and sentences. The theoretical inside of this paper is also weak. I think a good improvement direction is to discuss the relationship between the sampling error and the number of samples and how this paper can remediate the sampling error from the theoretical perspective. Some new interesting findings might also be discovered during this process.
3. This paper also introduces many fragile hyperparameters. For example, the learning rate for its proposed behavior policy ranges from 10^{-3} to 10^{-5} for an ADAM optimizer. Such a large difference in learning rate is not common for the ADAM optimizer. Moreover, they also require KL cutoff and regularizer coefficient inside the mini-PPO to encourage under-sampled data, which are two more dimensions of hyperparameters.

**Questions:**

What is the final choice of the batch size for PPO? Only candidates 1024, 2048, 4096, and 8192 are reported.

---

> ### Author Response · Authors · 2024-11-18
>
> Thank you for your review! We believe we can address your comments and questions by adding a few additional details on our hyperparameter choices in our revisions.
>
> # PPO batch size
> > This paper adjusts the batch size of PPO to "1024,2048,4096,8192" which is much larger than the original batch size (64 samples) of PPO...This drastic increase in batch size is not well-justified.
>
> Please note in our paper, the batch size refers to steps of environment interaction between target policy optimizations, not the mini-batch size used during optimization. **What we call “batch size” is called the “horizon (T)” in the original PPO paper, and they set this value to 2048 for MuJoCo experiments just as we do** (see Appendix A in [1]). PPO batch sizes between 1024 - 8192 are commonly used in MuJoCo tasks. For instance, the default batch size for PPO is 2048 in popular implementations of PPO in StableBaselines3, CleanRL, and Tianshou:
>
> * StableBaselines3: see Line 85 [here](https://github.com/DLR-RM/stable-baselines3/blob/e4f4f123e3b5afa828590b895ec22c7852872fe4/stable_baselines3/ppo/ppo.py#L85C7-L85C29)
> * CleanRL: see line 51 [here](https://github.com/vwxyzjn/cleanrl/blob/e648ee2dc8960c59ed3ee6caf9eb0c34b497958f/cleanrl/ppo_continuous_action.py#L51C1-L51C26)
> * Tianshou: see Line 34 [here](https://github.com/thu-ml/tianshou/blob/935a85a09fed1466379e26378c11821c6a7c9954/examples/mujoco/mujoco_ppo.py#L34)
>
> We sweep over batch sizes to ensure we accurately represent the performance of PPO with on-policy sampling, and we found that increasing the batch size beyond 1024 often led to large improvements in data efficiency. We attached a subset of our sweep over batch sizes [here](https://drive.google.com/file/d/1KEBV2p0OJGPpLVocDz-zlxf2UkD7olZG/view?usp=share_link) to support this claim (10 seeds per curve). For instance, on Hopper-v4, PPO achieves far greater returns with a batch size of 2048 compared to 1024.
>
> > In fact, it is confusing that their PPO algorithm still achieves the comparable performance of the original PPO with at least 16 times less update due to the batch size increase.
>
> The PPO paper uses the “-v1” versions of the MuJoCo tasks while we use the “-v4” versions, so our results are not directly comparable; the “-v1” versions use an old MuJoCo simulation engine and are outdated and no longer supported (see the Gymnasium documentation [here](https://gymnasium.farama.org/environments/mujoco/#versions)). Also note the original PPO paper uses different hyperparameters (e.g. they perform 3 epochs of policy updates whereas we perform 10).
>
> # Writing
>
> > The length of this paper can be greatly reduced. Currently, this paper has too many unnecessary examples and sentences.
>
> If there are examples and sentences that you believe could be omitted, could you please refer us to them? We could then or clarify their importance and further discuss whether they can be safely omitted or condensed.
>
> # Theory
>
> > The theoretical inside of this paper is also weak.
>
> Could you also please clarify the weaknesses in the existing theory we provide? We agree that it would be interesting to theoretically investigate how much PROPS reduces sampling as a function of the number of samples collected, though we believe our work is complete as is and appropriately supports our core claims: PROPS empirically reduces sampling error and improves data efficiency.
>
> # Learning rates with Adam optimizer
>
> > The learning rate for its proposed behavior policy ranges from 10^{-3} to 10^{-5} for an ADAM optimizer. Such a large difference in learning rate is not common for the ADAM optimizer.
>
> Our choice of learning rates with the Adam optimizer is consistent with what appears in the existing RL research. For instance, the default PPO learning rate in Stablebaselines3, CleanRL, and Tianshou is $3 * 10^{-4}$, and in TRL, a popular package for training LLMs, the default learning rate is $5 * 10^{-5}$:
>
> * StableBaselines3: see line 84 [here](https://github.com/DLR-RM/stable-baselines3/blob/e4f4f123e3b5afa828590b895ec22c7852872fe4/stable_baselines3/ppo/ppo.py#L84)
> * Tianshou: see line 30 [here](https://github.com/thu-ml/tianshou/blob/935a85a09fed1466379e26378c11821c6a7c9954/examples/mujoco/mujoco_ppo.py#L30)
> * CleanRL: see line 47 [here](https://github.com/vwxyzjn/cleanrl/blob/e648ee2dc8960c59ed3ee6caf9eb0c34b497958f/cleanrl/ppo_continuous_action.py#L47)
> * TRL: see the default `learning_rate` value in the PPOConfig in the documentation [here](https://huggingface.co/docs/trl/main/en/ppo_trainer#trl.PPOConfig).
>
> [1] Schulman et. al. Proximal Policy Optimization Algorithms. https://arxiv.org/pdf/1707.06347

---

> ### Author Response · Authors · 2024-11-18
>
> # Tuned PPO batch sizes used in our experiments
>
> > What is the final choice of the batch size for PPO? Only candidates 1024, 2048, 4096, and 8192 are reported.
>
> Below, we list the batch sizes we used in our PPO runs. We can include tuned hyperparameters in appendix with our revisions:
> * Swimmer-v4: 4096
> * HalfCheetah-v4: 1024
> * Hopper-v4: 2048
> * Walker2d-v4: 4096
> * Ant-v4: 1024
> * Humanoid: 8192
>
> ---
>
> Thank you again for your review! We believe our response clarifies your comments on hyper parameter choices, but if you have further questions, please ask! We'll be happy to provide further clarification.

---

> > ### Comment · Reviewer_8GpQ · 2024-11-26
> >
> > Thank you very much for your clarification. I still have a few concerns.
> >
> > Experiments:
> > > Swimmer-v4: 4096
> > HalfCheetah-v4: 1024
> > Hopper-v4: 2048
> > Walker2d-v4: 4096
> > Ant-v4: 1024
> > Humanoid: 8192
> >
> > 1. Batch size: For the reported batch size, I feel those numbers are not well-justified. For example, the Ant environment, with its 105-dimensional observation space, is more challenging compared to the Hopper and Walker environments, which have observation spaces of only 11 and 17 dimensions. However, the batch size for Ant is 1024 which is smaller than the batch size for Hopper and Walker. This is counterintuitive for machine learning applications.
> > The current approach in CleanRL is to set the batch size to be the same for all environments to show the generalizability of PPO algorithms. Therefore, in the modified PPO algorithms proposed by this paper, I do not think a collection of different batch sizes is convincing.
> >
> > 2. Learning rate: This paper searches the learning rate from $10^{-3}$ to $10^{-5}$. This is a fairly large range. I understand some LLM applications use $10^{-5}$ because their neural networks are much deeper and larger. However, the standard PPO for MuJoCo uses a neural network with only one 64*64 hidden layer. I do not think that the number $10^{-5}$ from LLM applications can be used to justify the hyperparameters here.
> > Moreover, this algorithm also introduces many new hyperparameters for their proposed PROPS algorithm.
> > ### PROPS Hyperparameters
> > - **Learning Rate:** \(10^{-3}, 10^{-4}\) (and \(10^{-5}\) for Swimmer)
> > - **Behavior Batch Size (m):** 256, 512, 1024, 2048, 4096 (satisfying \(m \leq n\))
> > - **KL Cutoff (\(\delta_{\text{PROPS}}\)):** 0.03, 0.05, 0.1
> > - **Regularizer Coefficient (\(\lambda\)):** 0.01, 0.1, 0.3
> >
> > I am not sure what would be the set of default parameters and what would be the performance of the set of the default parameter.
> >
> >
> > Theory: A question from my side is that
> >
> > 1. A smaller number of samples may have a mean further from the distribution mean. Thus, this scenario needs more "on-policy" sample correction. However, the smaller number of on-policy samples does not give enough space for PROPS to learn.
> >
> > 2. A larger number of samples may have a mean closer to the distribution mean. Thus, this scenario needs less sample "on-policy" sample correction and PROPS may not help much.
> >
> > This seems to be a paradox. It would be nice to have a theory to justify whether PROPS works on a small or large amount on-policy data and why.

---

> ### Author Response · Authors · 2024-12-04
>
> We appreciate the followup comments!
>
> # PPO batch sizes
>
> > For the reported batch size, I feel those numbers are not well-justified.
>
> We tuned our batch sizes to ensure we are comparing PROPS to the most competitive version of each algorithm. It is common in the literature to fix the batch size across all tasks in a given benchmark, though our approach is also valid.
>
> > the Ant environment, with its 105-dimensional observation space, is more challenging compared to the Hopper and Walker environments, which have observation spaces of only 11 and 17 dimensions. However, the batch size for Ant is 1024 which is smaller than the batch size for Hopper and Walker.
>
> Understanding why PPO can solve Ant-v4 with a batch size of 1024 is beyond the scope of this work. However, it’s worth noting that the Ant-v4 has only 27 state dimensions; the newer Ant-v5 state has 107 dimensions because it include contact forces (which are excluded in v4, see the [Gymnasium version history](https://gymnasium.farama.org/environments/mujoco/ant/#version-history)).
>
> # PROPS Hyperparameters
>
> > this algorithm also introduces many new hyperparameters for their proposed PROPS algorithm.
>
> [Figure: PROPS performance aggregated over a hyper parameter sweep.](https://drive.google.com/file/d/1s9KVUFWhYJJkqJMnK7qQd7rriF65FqWv/view?usp=share_link)
>
> In our initial response, we think we overlooked the core comment you made regarding hyperparameter sensitivity. PROPS admittedly introduces additional hyperparameters, but PROPS is not fragile to hyperparameter choices. In the figure linked above, we plot PROPS performance aggregated over the following PROPS hyperparameters from our hyperparameter sweep (we call this “PROPS (aggregate)” in the legend):
>
> * Learning Rate: $10^{-3}, 10^{-4}$ (and $10^{-5}$ for Swimmer)
> * KL Cutoff ($\delta_{\text{PROPS}}$): 0.03, 0.05, 0.1
> * Regularizer Coefficient ($\lambda$): 0.01, 0.1, 0.3
>
> The PROPS (aggregate) curves closely follow the tuned PROPS curves, indicating that PROPS is not fragile to hyperparameter choices.
>
> We would also like to clarify that we considered PROPS learning rates in $\{ 10^{-3}, 10^{-4}}$, and then additionally considered $10^{-5}$ for Swimmer-v4 *only* (we note this in Table 2). We considered PPO learning rates in $\{ 10^{-3}, 10^{-4} \}$, which is standard in the RL literature for MuJoCo tasks.
>
>
> > I am not sure what would be the set of default parameters and what would be the performance of the set of the default parameter.
>
> We suggest the following default hyperparameters:
> * Learning rate: 10^{-3}
> * Behavior batch size: 256
> * KL regularizer (lambda): 0.1
> * KL cutoff: 0.05
> We’ve added tuned PROPS hyperparameters for all environments in Table 4 of our revised submission, and these suggested default values are the most common values we saw in our tuned hyperparameters.
>
> # PROPS with small/large batch sizes
>
> > A smaller number of samples may have a mean further from the distribution mean. Thus, this scenario needs more "on-policy" sample correction. However, the smaller number of on-policy samples does not give enough space for PROPS to learn.
>
> PROPS can decrease sampling error even if there are only a few samples in the agent’s buffer. PROPS decreases the probability of sampling actions in the agent’s buffer (thus increasing the probability of sampling under-sampled actions), and if there are only a few samples in the buffer, this simply means that fewer actions will be pushed down (and thus the behavior policy may not change significantly).
>
> We’ll use a single N-arm bandit problem to make this point more concrete. Suppose the agent’s policy places 1/N probability on each arm and the agent’s buffer contains a single sample corresponding to arm 1. To reduce sampling error in the next step, the behavior policy should decrease the probability of sampling arm 1 (since arms 2 through N are currently under-sampled).
>
> > A larger number of samples may have a mean closer to the distribution mean. Thus, this scenario needs less sample "on-policy" sample correction and PROPS may not help much.
>
> Your reasoning here is correct; when there is a lot of data in the agent’s buffer, the data’s distribution may closely match the expected on-policy distribution, leaving little room for improvement with PROPS. Our experiments focus on settings where there is only a moderate amount of data in the agent’s buffer and thus sufficient room for improvement with PROPS.

---

### Public Comment · ~Zhiwei_Jia1 · 2024-11-18
**An interesting approach for improving on-policy policy-based RL**

Hi authors,

Thanks for the great work. Please consider citing GSL [1,2] in the related work section as GSL leverages off-policy sampling (from population-based specialist agents) for an on-policy agent (generalist), a framework that shares a similar spirit here.

[1] Improving Policy Optimization with Generalist-Specialist Learning, ICML 2022
[2] UniDexGrasp++: Improving Dexterous Grasping Policy Learning via Geometry-aware Curriculum and Iterative Generalist-Specialist Learning, ICCV 2023

---

### Author Response · Authors · 2024-12-04
**Revisions to our submission**

Dear Reviewers,

Thank you for taking the time to review this paper and for participating in discussion! We’re pleased to see the reviewers found our paper well-written `[iwvF, uFws, gv5j]` and of importance to the RL community  `[uFws, gv5j]`. We’ve uploaded a revised submission and highlighted important revisions in purple. We also point out the line numbers of our revisions in our summary below.

* `[8GpQ]`: We have clarified their comments regarding hyperparameter choices, added tuned hyperparameters to Table 3 and 4 of Appendix E, and showed in our rebuttal that PROPS is not fragile to hyperparameters.
* `[uFws]`: This reviewer verified they understand our PROPS method much better after reading our rebuttal.  In our revisions, we’ve clarified the importance of PROPS’s KL regularization in continuous action tasks (footnote 3 on page 6) and more clearly show how the PROPS objective relates to the PPO objective (lines 297 - 307).
* `[gv5j]`: This reviewer was largely positive, and pointed out a few minor notational changes to us.
* `[iwvF]`: We have clarified comments about our method’s empirical findings, choice of baselines, and interplay between different hyperparameters. In our revisions, we’ve added additional theory to Appendix A to explain the relationship between hyperparameters  (lines 767 - 844).

---

### Meta-Review · Area_Chair_1Eji · 2024-12-21

**Metareview:**

This paper introduces an adaptive sampling method aimed at reducing the distribution mismatch between the empirical data distribution in the buffer and the target policy. The problem addressed is both interesting and underexplored in existing literature. However, the primary concerns raised by reviewers focus on the experimental evaluation. Specifically, given that the work centers on improving the sample efficiency of policy gradient methods through off-policy sampling, it should include comparisons with established off-policy methods. Furthermore, the improvement over PPO appears quite marginal, especially considering the additional algorithmic complexity introduced.
On a minor note, the theoretical contribution, particularly Proposition 1, seems trivial and appears very loosely connected to the subsequent algorithmic design.

**Additional Comments On Reviewer Discussion:**

The primary concerns raised by reviewers focus on the experimental evaluation. Specifically, given that the work centers on improving the sample efficiency of policy gradient methods through off-policy sampling, it should include comparisons with established off-policy methods. Furthermore, the improvement over PPO appears quite marginal, especially considering the additional algorithmic complexity introduced. I feel these issues were not adequately addressed during the rebuttal phase, which is the main reason I lean toward rejection.

---

### Decision · Program_Chairs · 2025-01-22

Reject